# Structural basis of sequence-specific cytosine deamination by double-stranded DNA deaminase toxin DddA

Lulu Yin[1,2,3], Ke Shi[1,2,3] & Hideki Aihara [1,2,3] ✉

The interbacterial deaminase toxin DddA catalyzes cytosine-to-uracil conversion in double-stranded (ds) DNA and enables CRISPR-free mitochondrial base editing, but the molecular mechanisms underlying its unique substrate selectivity have remained elusive. Here, we report crystal structures of DddA bound to a dsDNA substrate containing the 5′-TC target motif. These structures show that DddA binds to the minor groove of a sharply bent dsDNA and engages the target cytosine extruded from the double helix. DddA Phe1375 intercalates in dsDNA and displaces the 5′ (−1) thymine, which in turn replaces the target (0) cytosine and forms a noncanonical T–G base pair with the juxtaposed guanine. This tandem displacement mechanism allows DddA to locate a target cytosine without flipping it into the active site. Biochemical experiments demonstrate that DNA base mismatches enhance the DddA deaminase activity and relax its sequence selectivity. On the basis of the structural information, we further identified DddA mutants that exhibit attenuated activity or altered substrate preference. Our studies may help design new tools useful in genome editing or other applications.

Enzymatic deamination of cytosines in DNA plays key roles in various important biological processes, including innate immune responses against viruses and transposons, antibody diversification in adaptive immunity and the accumulation of somatic mutations in various human cancers[1–4]. The activity of APOBEC family single-stranded (ss) DNA cytosine deaminases has also been harnessed in base-editing technologies, where an engineered Cas9-guide RNA complex directs APOBECs for site-specific C-to-T base substitutions in genomic DNA without making double-strand breaks[5]. Cytosine deamination by the APOBEC enzymes is sequence selective; for instance, human APOBEC3A (A3A) and APOBEC3B (A3B) only deaminate cytosines in a 5′-T**C** sequence context (deaminated C is in bold), which is responsible for the characteristic 'APOBEC signature' mutations found widely in cancer genomes[6,7]. Structural studies have shown that A3A and A3B bind ssDNA substrates in a U-shaped conformation, with the thymine base 5′ (−1) to the target cytosine flipped out and making specific contacts with the protein[8]. A

similar mode of hairpin-shaped substrate engagement was observed for a distantly related bacterial transfer RNA adenosine deaminase, TadA, which served as the template for an evolved DNA adenine deaminase capable of A-to-G conversion in base editing[9,10].

Recent studies have identified a dsDNA deaminase from *Burkholderia cenocepacia*, DddA, an interbacterial toxin that is delivered to contacting cells by the type VI secretion system and mediates antagonism between Gram-negative bacteria[11,12]. Interestingly, DddA shares a strong preference for the 5′-T**C** target sequence with A3A, A3B and several other APOBEC family members[12]. However, unlike APOBECs that only deaminate ssDNA, DddA selectively deaminates cytosines in dsDNA. The unique activity of DddA allowed Mok et al. to develop CRISPR-free DddA-derived cytosine base editors, which enable C-to-T base editing in mitochondrial, chloroplast and nuclear DNA[12–20]. Furthermore, Cho et al. showed that a catalytically inactive DddA mutant (E1347A) fused to the TadA-derived DNA adenine deaminase mediates

[1]Department of Biochemistry, Molecular Biology, and Biophysics, University of Minnesota, Minneapolis, MN, USA. [2]Institute for Molecular Virology, University of Minnesota, Minneapolis, MN, USA. [3]Masonic Cancer Center, University of Minnesota, Minneapolis, MN, USA. ✉e-mail: aihar001@umn.edu

targeted A-to-G editing in human mitochondrial DNA, where DddA may assist in unwinding/melting of the dsDNA substrate[21]. In addition, DddA has been adapted by Gallagher et al. for genome-wide protein–DNA interaction site mapping in bacteria[22]. However, despite its useful applications, molecular mechanisms underlying the biochemical activities of DddA have remained unknown. Here we report crystal structures of DddA in complex with dsDNA and corroborating biochemical data, which together reveal a unique mechanism of substrate DNA recognition of DddA.

## Results

### Overall structure of DddA–dsDNA complex

To understand how DddA interacts with dsDNA substrates, we crystallized the toxin domain (Gly1290 to Pro1422) of *B. cenocepacia* DddA in complex with a 14-base pair (bp) dsDNA substrate containing the 5′-T**C** target sequence (Fig. 1a,b). DddA with a substitution of the catalytically essential glutamic acid residue (E1347A) was used to capture the enzyme–substrate complex. The structure of the DddA–dsDNA complex was determined in two different crystal forms and refined to 2.39 and 2.62 Å resolution, respectively (Table 1). The crystal structures show that DddA engages the minor groove of a sharply bent dsDNA (Fig. 1c,d). The structures obtained in the two crystal forms are very similar overall, with a root mean square deviation (r.m.s.d.) of 1.37 Å for all protein and DNA atoms, and of 0.45 Å for the protein backbone atoms, although they differ in the conformation of the target (0) 2′-deoxycytidine nucleotide. In the first structure (PDB 8E5E), the target cytosine base is completely flipped out of the DNA double helix and captured in the active site pocket, where it interacts with the Zn ion (Fig. 2a,c and Extended Data Figs. 1 and 2). In the second structure (PDB 8E5D), the target cytosine is parked in the major groove via a T-shaped stacking on the edge of the adjacent (+1) cytosine base, and the active site pocket is occupied by a phosphate ion (Fig. 2b,c and Extended Data Fig. 1). In both structures, the dsDNA substrate bound by DddA is bent away from the protein by ~80°, which leads to a substantially widened minor groove (groove width up to 15 Å, in comparison to 6 Å in the B-form DNA; calculated using CURVES+)[23], allowing for direct base contacts by the protein. Correspondingly, several nucleotides surrounding the 5′-⁻¹T**C**⁰ motif, including G (−2) and C (+1) of the deaminated strand and A (−1) of the complementary strand (unpaired due to the shift of −1 T; see below), show the A-form-like C3′-endo sugar pucker in both structures.

The structure of the Zn-dependent deaminase fold of DddA in complex with DNA shows minimal changes from that in complex with the immunity protein DddI (PDB 6U08)[12], with an overall backbone r.m.s.d. of 0.50 and 0.62 Å, respectively, for the two DNA-bound structures. A structural comparison highlights DNA mimicry by DddI in blocking the active site of DddA (Extended Data Fig. 3). Besides the active site zinc ion, in both our DddA–dsDNA structures we observed electron density for a putative metal ion octahedrally coordinated by the backbone carbonyl oxygen of Glu1381, Thr1382, Leu1384 and Asn1417, and both the backbone and side chain oxygen atoms of Asn1415. This density was modeled as a magnesium ion, which appears to stabilize the DddA residues important for DNA binding (Extended Data Fig. 4). Biochemical experiment showed that although the bound magnesium ion is not essential it enhances DddA deaminase activity (Extended Data Fig. 5), which is consistent with its structural role.

### Mechanism of TC motif recognition

The minor groove interaction by DddA is centered on Phe1375, which intercalates in dsDNA and displaces thymine at −1 position (5′ to the target cytosine) (Fig. 1c,d and Extended Data Fig. 1). The displaced thymine in turn replaces the target (0) cytosine extruded from the double helix (Fig. 3a). This unique arrangement is stabilized by bifurcated hydrogen bonds donated to the thymine O4 atom from the juxtaposed guanine base N1 and N2 atoms (Fig. 3b). His1345, which is one of the

**Table 1 | X-ray data collection and model refinement statistics**

|  | Crystal form 1 (PDB 8E5E) | Crystal form 2 (PDB 8E5D) |
|---|---|---|
| **Data collection** |  |  |
| Resolution range (Å) | 54.34–2.62 (2.74–2.62) | 44.77–2.39 (2.55–2.39)ᵃ |
| Space group | $P6_122$ | $P222_1$ |
| Unit cell |  |  |
| *a*, *b*, *c* (Å) | 62.95, 62.95, 237.08 | 31.70, 94.61, 138.47 |
| Total reflections | 50,693 (7,585) | 67,523 (4,225) |
| Unique reflections | 8,887 (1,076) | 13,177 (659) |
| Multiplicity | 5.7 (7.0) | 5.1 (6.4) |
| Completeness (%, spherical) | 98.0 (99.9) | 76.4 (23.4) |
| Completeness (%, ellipsoidal) |  | 93.2 (83.4) |
| $\langle I/\sigma(I)\rangle$ | 45.6 (1.4) | 8.7 (1.8) |
| $R_{merge}$ | 0.117 (1.647) | 0.092 (1.139) |
| $R_{meas}$ | 0.145 (1.911) | 0.101 (1.240) |
| $R_{pim}$ | 0.084 (0.955) | 0.042 (0.485) |
| $CC_{1/2}$ | 0.979 (0.589) | 0.998 (0.804) |
| **Refinement** |  |  |
| Reflections for $R_{work}$ | 8,863 (855) | 13,175 (346) |
| Reflections for $R_{free}$ | 458 (45) | 667 (21) |
| $R_{work}/R_{free}$ | 0.242/0.258 | 0.198/0.233 |
| Non-H atoms | 1,556 | 1,591 |
| Macromolecules | 1,552 | 1,572 |
| Ligands | 2 | 7 |
| Solvent | 2 | 12 |
| Average *B* factor (Å²) | 111.83 | 54.99 |
| Macromolecules | 94.82 | 45.04 |
| Ligands | 97.64 | 47.53 |
| Solvent | 89.25 | 40.25 |
| R.m.s. deviations |  |  |
| Bond lengths (Å) | 0.003 | 0.005 |
| Bond angles (°) | 0.53 | 0.78 |

Statistics for the highest resolution shell are shown in parentheses. ᵃFinal resolution cutoff was 2.39 Å along *a**, 2.74 Å along *b** and 2.70 Å along *c**, for anisotropic diffraction of the crystal.

Zn-coordinating residues, also donates a hydrogen bond to the thymine O2 atom. Thus, the strong 5′-T**C** preference of DddA appears to reflect the favorable interaction made by the −1 T base in replacing the target cytosine in the double helix. The noncanonical T–G interaction, which is distinct from the G•T wobble pair commonly observed in RNA secondary structures, is further stabilized by van der Waals contacts made by Ala1341 and a hydrogen bond between the carbonyl oxygen of Pro1338 and the guanine base N2 atom (Extended Data Fig. 6). Met1379 complements Phe1375 and Ala1341 to form a cluster of hydrophobic side chains inserted into the minor groove, interacting with the orphan (unpaired) adenine at the −1 position and stabilizing unstacked bases of dsDNA in the distorted conformation (Figs. 1d and 3a). Upstream of the 5′-T**C** motif, Asn1378 and Arg1403 are inserted into the DNA minor groove and interact with guanine at the position −2 of the deaminated strand and thymine at −4 of the complementary strand, respectively,

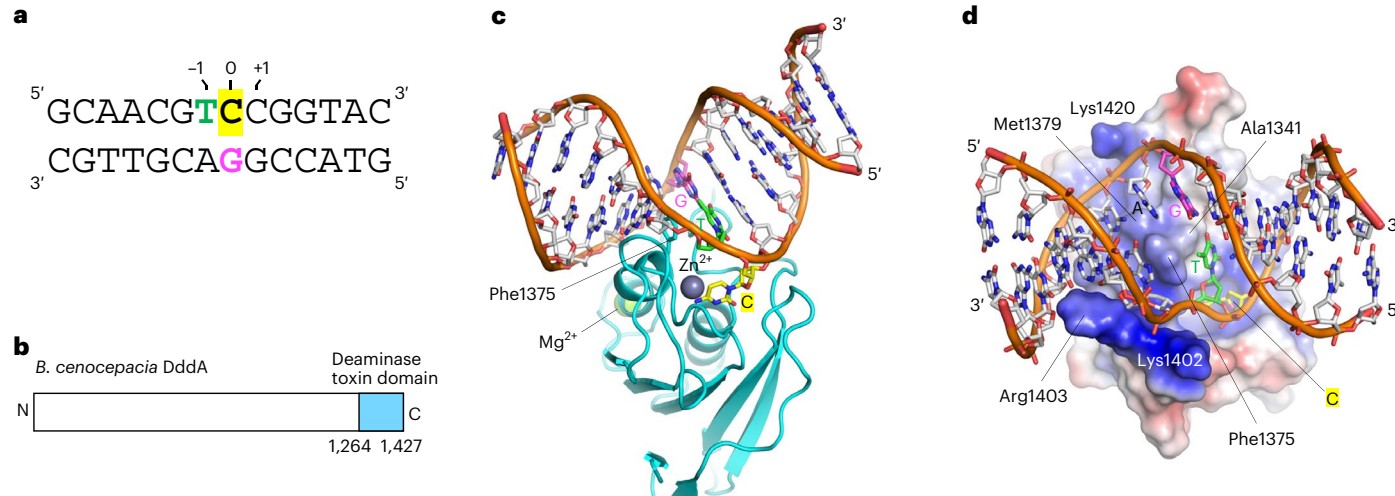

**Fig. 1 | Crystal structure of the DddA–dsDNA complex. a**, Sequence of the 14 bp DNA substrate used in our crystallographic studies, designed based on the sequence preference of DddA in *Escherichia coli* reported previously[12]. **b**, Schematic diagram showing the location of the deaminase toxin domain in full-length *B. cenocepacia* DddA. **c**, Overall view of the DddA–dsDNA complex, with the target cytosine flipped out of the double helix and engaged in the enzyme active site. The color scheme for nucleotides at the −1 and 0th positions follows that in **a**. **d**, An alternative view of the DddA–dsDNA complex, with the DddA molecular surface colored according to electrostatic potential (−2.5*kT/e* in red to +2.5*kT/e* in blue) as calculated by APBS[36].

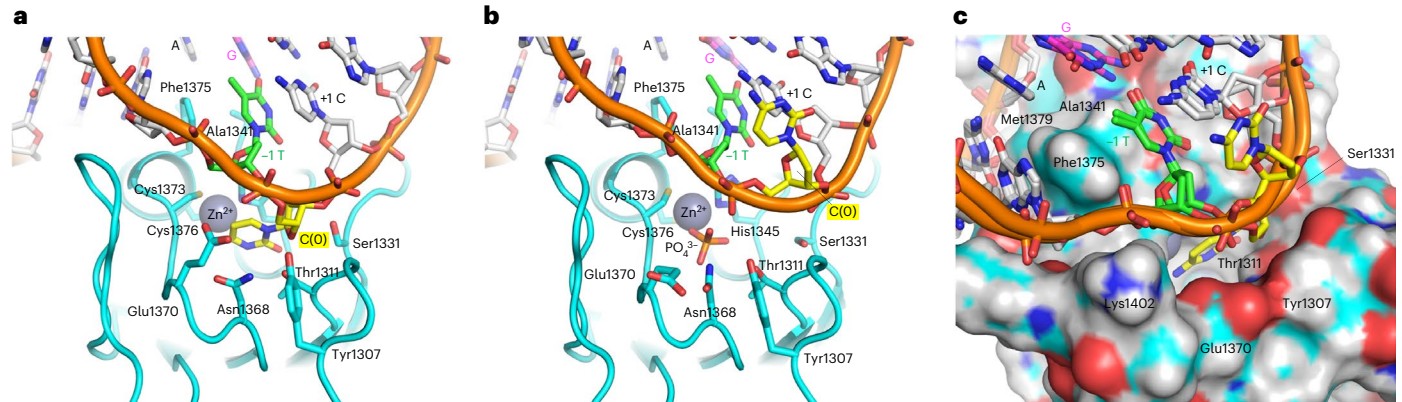

**Fig. 2 | Alternative conformations of the target (0) cytosine. a**, Structure of the DddA–dsDNA complex as in Fig. 1c, with the target cytosine engaged in the enzyme active site pocket. **b**, Structure of the DddA–dsDNA complex in an alternative conformation, in which the extrahelical target cytosine is stacked on the +1 cytosine base. **c**, Superposition of the two different DNA conformations. DddA protein surface is shown for the structure with the target cytosine in the active site pocket, as in **a**.

which may modestly contribute to sequence preferences (Extended Data Fig. 6). Binding of DddA to the bent DNA is also supported by interaction with the backbone phosphate groups from both strands, involving residues Ser1331, Asn1339, Tyr1340, Lys1402 and Lys1420 (Extended Data Fig. 6).

### Base mismatches promote DddA activity

On the basis of the highly distorted conformation of the dsDNA bound to DddA, we reasoned that base mismatches at the target (0) or 5′ (−1) position would destabilize the double helical structure of the substrate and facilitate DNA deamination by DddA. Thus, we compared DddA activity on fully base paired, singly mismatched (at position 0), and doubly mismatched (at positions 0 and −1) 14-bp dsDNA substrates (Fig. 4a,b). DddA deaminates cytosine in the 5′-T**C** motif in the fully base-paired substrate, in which the complementary strand has opposing 5′-GA (Fig. 4b, lane 6). Using a complementary strand with a single mismatch (5′-TA) led to enhanced activity, confirming our hypothesis (Fig. 4b, lane 4). The deamination reaction was even more efficient

with a complementary strand with double mismatches (5′-TT), consistent with our structural observation that substrate engagement by DddA requires disruption of base pairs at both positions 0 and −1 (Fig. 4b, lane 5).

Next, we further hypothesized that base mismatches may relax the 5′-T**C** requirement of DddA and examined whether DddA can deaminate cytosines preceded by different −1 bases (5′-G**C**, 5′-C**C**, 5′-A**C**) when paired with mismatched complementary strands (Fig. 4c). For the original complementary strand with 5′-GA, which would generate mismatches at the −1 position, we observed DddA-mediated deamination on all three substrates to a varying extent; the activity was highest on 5′-A**C** and poor on 5′-G**C** (Fig. 4c, lanes 9–11). For the complementary strand with opposing 5′-TT, we also observed deamination on all three substrates but their preferences were reversed; the activity was highest on 5′-G**C** and modest on 5′-A**C**, which forms a single mismatch at position 0 (Fig. 4c, lanes 6–8). With opposing 5′-TA, the activity was high on all three doubly mismatched substrates (Fig. 4c, lanes 3–5). Of note, the 5′-C**C** target was deaminated at both (−1 and 0) cytosines,

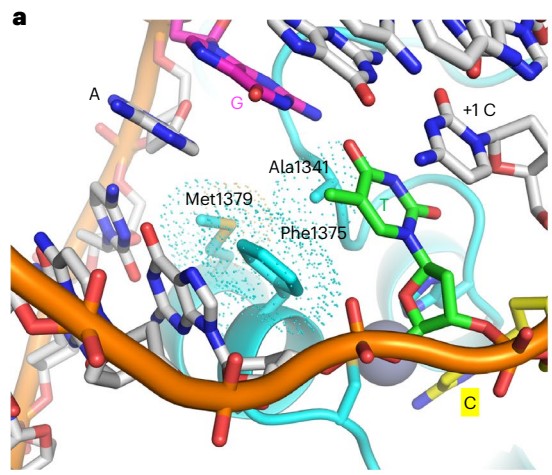

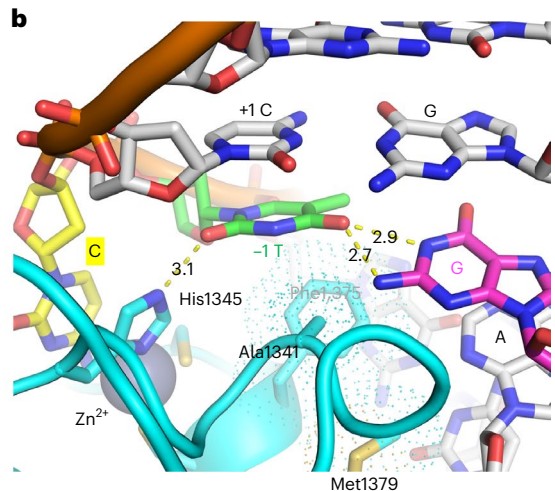

**Fig. 3 | Structural basis of 5′-TC target recognition. a**, DNA minor groove interaction by DddA centered on a cluster of hydrophobic residues (Ala1341, Phe1375, Met1379). The −1 thymine base in green is displaced by Phe1375, which in turn replaces the extrahelical target (0) cytosine. **b**, The −1 thymine base in the shifted register, stabilized by hydrogen bonds (yellow dashed lines with distances shown, in angstroms) to the juxtaposed guanine base and a zinc-coordinating residue His1345. Van der Waals radii for Ala1341, Phe1375 and Met1379 are indicated by dots.

which was confirmed by testing substrates labeled at either the 5′ or 3′ terminus of the target strand (Fig. 4c, lanes 4, 7 and 10, and Extended Data Fig. 7). These results show that base mismatches at either position 0 or −1 eliminate the 5′-T**C** requirement of DddA, although the sequence context matters in some cases.

## DddA mutants

To dissect structure–function relationships, we explored amino acid substitutions for key DNA-interacting residues of DddA (Fig. 5a and Supplementary Fig. 1). As mentioned above, a triad of hydrophobic residues, Ala1341, Phe1375 and Met1379, support unstacked bases of DddA-bound dsDNA in the minor groove (Figs. 1 and 3a,b). For Ala1341, which abuts against the noncanonical T−G base pair, we tested substitutions of Ser, Thr, Glu, Tyr and Pro. Of these mutants, only DddA A1341P retained activity on the canonical substrate (5′-T**C**/GA), and it showed the 5′-T**C** preference (Fig. 5a,b). Interestingly, although the activity of DddA A1341P on the fully base-paired substrate was weaker than that of the wild type, DddA A1341P showed higher activities than the wild type on all mismatch-containing substrates (Fig. 5c,d, compare with Fig. 4b,c). The hydrophobic proline side chain inserted more deeply (than alanine) into the minor groove may interact favorably with unpaired DNA bases. For the DNA-intercalating residue Phe1375, either Ala (F1375A) or Arg (F1375R) substitution led to a complete loss of the deaminase activity, while a variant with Tyr substitution (F1375Y) showed residual activity, which highlights the importance of the π-stacking interaction (Fig. 5a). DddA F1375Y also showed activities on mismatched substrates (Supplementary Fig. 1). For Met1379, either Ala (M1379A) or Arg (M1379R) substitution abolished the deaminase activity (Fig. 5a). These results show the importance of the hydrophobic patch of DddA in DNA substrate engagement and that structural perturbation of this region affects the target preference.

One of the DddA residues positioned near the DNA backbone is Glu1370, which forms a part of the rim of the deep active site pocket along with Tyr1307 (Fig. 2c). In the structure with the cytosine base parked in the DNA major groove, Glu1370 side chain is pointed away from the DNA (Fig. 2b). When the cytosine base is engaged in the active site pocket, Glu1370 appears to be oriented toward DNA with ~3.7 Å between the carboxyl and phosphate groups, although weak electron density suggests high flexibility of this side chain (Fig. 2a and Extended Data Fig. 1a). Substitution of either Lys (E1370K) or

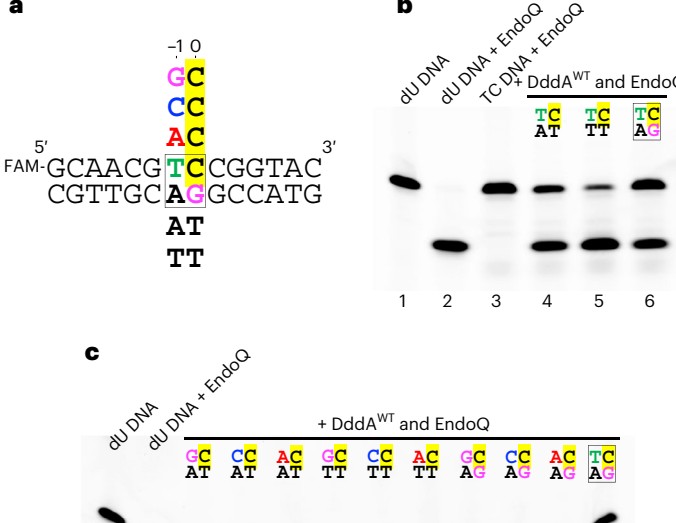

**Fig. 4 | Base mismatches enhance DddA activity and eliminate its 5′-TC requirement. a**, Sequence variation of the top and bottom strands used to generate mismatch-containing DNA substrates. FAM denotes fluorescein. **b**, Deamination by wild-type DddA of the T**C**-containing target strand, annealed to the fully base-paired (lane 6) or mismatched bottom strands (lanes 4 and 5). The top bands are uncleaved 14-mer substrate DNAs, whereas the bottom bands are deamination products subsequently cleaved by the lesion-specific endonuclease pfuEndoQ[29]. The control 'dU DNA' contains 2′-deoxyuridine in place of the target C. Representative result of ten replicates is shown. **c**, Deamination by wild-type DddA of the non-T**C** target strands, annealed to mismatched bottom strands. The top and bottom strand sequences for the −1 and 0th positions are shown above each lane. Lane 12 shows a reaction on the canonical substrate (fully base-paired T**C** target) for reference. Representative result of three replicates is shown. The 5′ fluorescein-labeled substrates were used in all experiments shown in this figure.

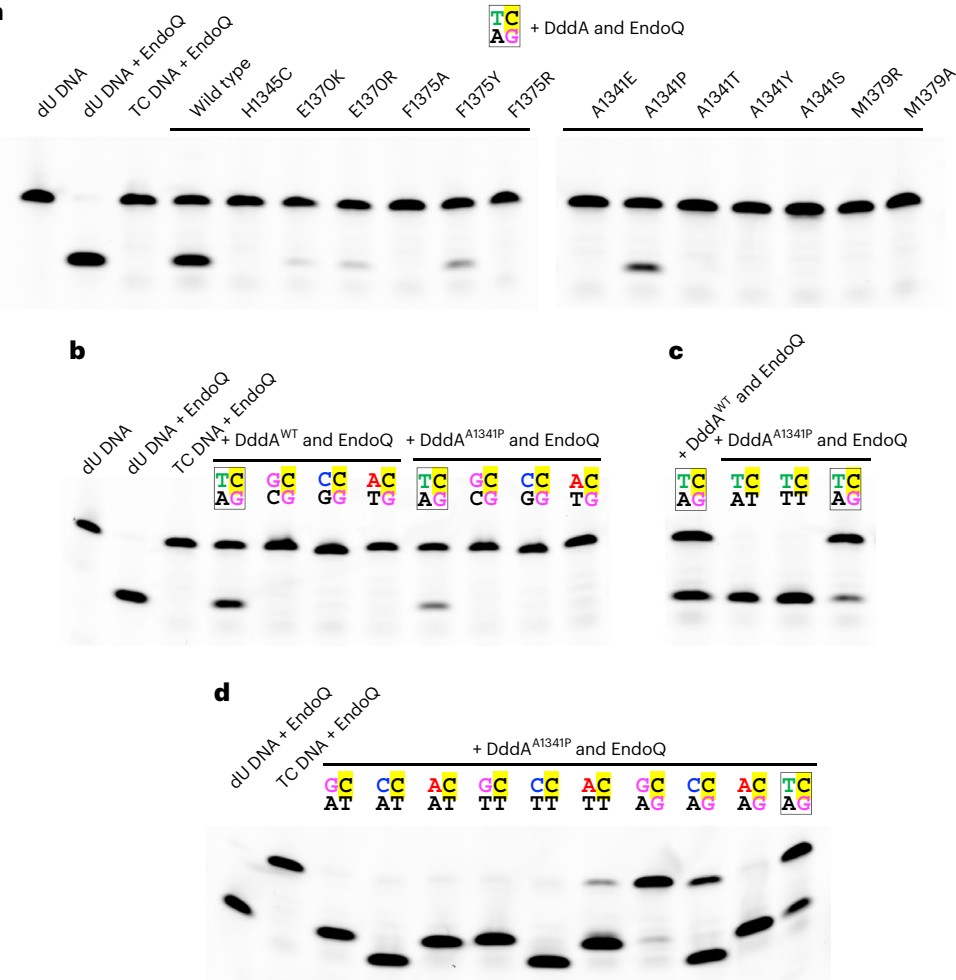

**Fig. 5 | DddA mutants. a**, Activities of DddA point mutants on the canonical (fully base-paired T**C** target) substrate. **b**, A comparison between wild type and A1341P on the fully base-paired DNA substrates with various −1 bases. **c**, Activities of DddA A1341P on the T**C**-containing target strand annealed to the fully base-paired or mismatched bottom strands. **d**, Deamination by DddA A1341P of the non-T**C** target strands annealed to mismatched bottom strands (the same set of substrates as in Fig. 4c). The 5′ fluorescein-labeled substrates were used in all experiments in this figure. Gels are representative of two replicates.

Arg (E1370R) for Glu1370, which installs a positive charge to interact favorably with the DNA backbone phosphate, made DddA less active than the wild type (Fig. 5a and Supplementary Fig. 1). It is possible that the dynamics of this residue plays a role in flipping the target cytosine base into the active site. Lastly, replacing His1345 with Cys, an alternative Zn-coordinating residue as found in some cytidine deaminases[24], abolished the DddA activity (Fig. 5a).

## Discussion

Our structural studies show that DddA active site captures the target cytosine base that has completely swung out of the DNA double helix (Figs. 1 and 2a,c). Similar base-flipping mechanisms have been observed for various nucleic acid repair or modifying enzymes, including DNA glycosylase, cytosine methyltransferase, dsRNA adenosine deaminase and lesion-specific endonuclease[25–32]. A hallmark feature of these enzymes is the intercalation of an amino acid side chain into DNA/RNA base stacks to fill a void in the double helix[33]. Another frequently observed feature is a sharp kink in the dsDNA substrate with unstacked bases, which also facilitates base flipping[25,29,31,34]. DddA uses both these strategies—the dsDNA bound by DddA is sharply bent at the base step 5′ to the 5′-T**C** motif, and Phe1375 inserts deeply into the minor groove. However, the mechanism of base flipping by DddA is distinct in that the intercalated phenylalanine replaces the adjacent (−1) thymine rather than the target (0) cytosine base itself (Fig. 3). This unique arrangement causes tandem displacement and a shift in the register of base pairing, with the target cytosine base extruded from the double helix. The DddA–dsDNA structure trapped with the target cytosine parked in the major groove (Fig. 2b and Extended Data Fig. 1b) suggests that DddA can locate 5′-T**C** motifs in double-stranded DNA without engaging the cytosine base in the active site. It may represent an intermediate conformation that allows DddA to scan through a DNA sequence to locate target cytosines.

The mechanism of 5′-T**C** target recognition by DddA is distinct from that of APOBEC family ssDNA deaminases (Extended Data Fig. 8). We showed previously that ssDNA substrates bound to A3A and A3B take a U-shaped conformation with the −1 thymine base bound in a groove on the enzyme surface, where it forms hydrogen bonds with a key Asp side chain[8]. In contrast, the −1 thymine in dsDNA bound to DddA remains intrahelical and is paired with a guanine base, where it makes both DNA base (guanine) and protein side chain (His1345) contacts (Fig. 3b). The hydrogen bonding to a DNA base in the widened minor groove by Zn-coordinating His1345 of DddA is distinct from the shape readout mechanism through histidine insertion into a compressed minor groove used by various DNA-binding proteins[35]. The strong 5′-T**C** selectivity of DddA suggests that the noncanonical T−G interaction is required for the target cytosine base flipping, which is corroborated

by the dramatically relaxed target sequence selectivity of DddA on mismatch-containing dsDNA substrates. Residual sequence dependence observed for the mismatched substrates (for example, Fig. 4c, lane 9 versus lane 11) may reflect how efficiently the −1 base replaces the target (0) cytosine by interacting with its juxtaposed base and the surrounding protein residues, including His1345, in the distorted dsDNA conformation.

While most amino acid substitutions that affect the key DNA minor groove interaction of DddA led to a loss of the enzymatic activity, several mutant enzymes retained DNA deaminase activity (Fig. 5 and Supplementary Fig. 1). These attenuated DddA variants could be useful in reducing off-target mutations or alleviating cytotoxicity in base editing, as shown in recent studies[16,19]. In addition, the enhanced activity of DddA A1341P toward mismatch-containing substrates (Fig. 4c compared with Fig. 5d) suggests that it might be possible to engineer DddA to expand its targets. In this context, it is notable that recent directed evolution experiments have identified DddA11, a DddA variant containing A1341V and E1370K amino acid substitutions, which can edit non-TC targets in both mitochondrial and nuclear DNA[18]. Our studies reported here will be instrumental in further structure-based engineering of DddA for base editing or other new applications, either as the deaminase catalytic component or a vehicle for other DNA-modifying enzymes.

## Online content

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

## Methods

### Protein expression and purification

DddA(1290–1422) with an E1347A deaminase-inactivating amino acid substitution was expressed in *E. coli* strain BL21(DE3) using the pET-24a vector with a C-terminal 6xHis-tag and an additional methionine on the N terminus. Transformed *E. coli* was grown in 4 liters LB medium with 40 mg l$^{-1}$ kanamycin at 37 °C until the optical density (OD$_{600}$) reached 0.8, at which point the protein expression was induced by adding IPTG and ZnCl$_2$ to the final concentrations of 0.5 mM and 50 μM, respectively. After overnight shaking and incubation at 18 °C, the bacteria were pelleted; resuspended in 20 mM Tris–HCl, pH 7.4, 0.5 M NaCl, 5 mM β-mercaptoethanol and 5 mM imidazole; and lysed by sonication in the presence of hen egg-white lysozyme (0.4 mg ml$^{-1}$). The lysate was cleared by centrifugation at 64,000*g* for 1 h at 4 °C, filtered through a 0.2-μm asymmetric polyethersulfone (aPES) membrane and applied to a 5-ml Ni-NTA Superflow cartridge (QIAGEN). After extensive washing of the cartridge with the same buffer as above, bound protein was eluted with a linear gradient of imidazole from 5 to 300 mM over 165 ml. Fractions containing DddA(1290–1422)-E1347A were identified by SDS–PAGE, concentrated to 5 ml using an Amicon Ultra-15 centrifugal filter, 3 kDa molecular weight cutoff (MWCO) (Millipore Sigma), and further purified by size-exclusion chromatography (SEC) on a HiLoad 26/600 Superdex 75 column. An N-terminal fragment, DddA(1290–1396), was expressed as MBP-fusion using the pMAL-c5x vector with an 8xHis-tag and an HRV 3C protease cleavage site between MBP and DddA. The wild type or various mutant derivatives of this fusion protein were expressed and purified as above, except that the MBP-His$_8$ tag was cleaved after the nickel affinity step by overnight incubation with HRV 3C protease. Purified proteins were concentrated by ultrafiltration using Amicon centrifugal filters in the SEC buffer containing 20 mM Tris–HCl, pH 7.4, 0.5 M NaCl and 5 mM β-mercaptoethanol; flash-frozen in liquid nitrogen; and stored at −80 °C. Protein concentrations were determined on the basis of UV absorbance measured on a Nanodrop 8000 spectrophotometer. Mass spectrometry showed that DddA(1290–1422)-E1347A used in the crystallographic studies had lost the N-terminal methionine residue. Specifically for the experiment to investigate metal ion dependency (Extended Data Fig. 5), DddA(1290–1396) was purified with 1 mM EDTA included in the final SEC buffer (20 mM Tris–HCl, pH 7.4, 0.5 M NaCl, 0.5 mM tris(2-carboxyethyl)phosphine (TCEP) and 1 mM EDTA) to remove the bound Zn$^{2+}$ and Mg$^{2+}$ ions.

### Crystallization and structure determination

DddA(1290–1422)-E1347A at ~12 mg ml$^{-1}$ was mixed with 1.5× molar excess of a 14 bp dsDNA (5′-GCAACGT**C**CGGTAC/5′-GTACCGG**A**CGTTG C; the 5′-T**C** target motif is underlined) and dialyzed overnight at 4 °C against 10 mM Tris–HCl, pH 7.4, 0.1 M NaCl, 0.5 mM TCEP in a Slide-A-Lyzer MINI Dialysis Device, 2 kDa MWCO (Thermo Scientific). The dialyzed complex was subjected to crystallization screening without further concentration in the sitting-drop vapor diffusion mode at ambient temperature by mixing 0.1 μl each of the complex and reservoir solutions. We obtained crystals in two different conditions. Crystal form 1 obtained in (0.2 M magnesium chloride, 0.1 M Tris–HCl, pH 8.5, 25% polyethylene glycol 3350) yielded the structure with the target cytosine in the active site pocket at 2.62 Å resolution (PDB 8E5E). Crystal form 2 obtained in (0.2 M sodium dihydrogen phosphate, 20% polyethylene glycol 3350) yielded the structure with the target cytosine parked in the DNA major groove at 2.39 Å resolution (PDB 8E5D). The DddA–dsDNA crystals were cryo-protected by brief soaking in the respective reservoir solution supplemented with 20% ethylene glycol and flash cooled by plunging in liquid nitrogen. X-ray diffraction data were collected at the NE-CAT beamline 24-ID-C of the Advanced Photon Source (Lemont, IL). The 8E5E dataset was processed using DIALS (https://dials.github.io). The 8E5D dataset exhibited anisotropic diffraction, and the dataset was processed with autoPROC[37], which

implements XDS[38] for integration, followed by three other programs from CCP4 Suite[39]: POINTLESS[40], AIMLESS[41] and TRUNCATE[42] for reduction, scaling and structure factor calculation, respectively. Anisotropic diffraction analysis and truncation were done with STARANISO (https://staraniso.globalphasing.org/). The structures were determined by molecular replacement with PHASER[43] using the previously reported inhibitor (DddI)-bound DddA structure[12] (PDB 6U08) as the search model. Iterative model building and refinement were conducted using Coot[44] and PHENIX[45]. The final resolution cutoffs for both crystal structures were determined by paired refinement[46] (Extended Data Fig. 9). A summary of crystallographic data statistics is shown in Table 1. Figures were generated using PyMOL (https://pymol.org/2/).

### DddA activity assay

To reconstitute the active enzyme, DddA(1290–1396) was mixed with 10× molar excess of a chemically synthesized and HPLC-purified (purity >90%, BIOMATIK) C-terminal peptide corresponding to the residues 1397–1422 (GAIPVKRGATGETKVFTGNSNSPKSP). The deaminase assay was conducted with a 5′-fluorescein-labeled 14-mer DNA oligonucleotide (5′-GCAACG**TC**CGGTAC-3′) or its variants with different −1 bases (5′-G**C**, 5′-C**C**, 5′-A**C**), annealed to an unlabeled 14-mer complementary DNA strand (5′-GTACCGG**GA**CGTTGC) or its variants with 5′-TT, 5′-TA, 5′-GC, 5′-GG or 5′-GT in place of the underlined 5′-GA. The reactions contained 200 nM dsDNA substrate, 10 μM DddA(1290–1396), 100 μM DddA(1397–1422), 40 mM Tris–HCl, pH 7.4, 50 mM KCl, 1.0 mM MgCl$_2$, 1.0 mM dithiothreitol. Following incubation at 37 °C for 50 min, pfuEndoQ was added to the final concentration of 1.0 μM and the samples were further incubated at 60 °C for 30 min to cleave deaminated products[29]. The reactions were stopped by the addition of formamide to 65% and heating to 95 °C for 10 min. The products were separated by gel electrophoresis on a 15% polyacrylamide TBE–urea denaturing gel and visualized by scanning on a Typhoon FLA 9500 imager. For every experiment, the activity of pfuEndoQ was verified on a control DNA oligonucleotide with dU (2′-deoxyuridine) in place of the target C. Specifically in the experiment shown in Extended Data Fig. 7, 3′-fluorescein-labeled DNA substrates were used. All oligonucleotides were obtained from Integrated DNA Technologies.

For investigating metal ion dependency, DddA(1290–1396) purified in the presence of 1 mM EDTA was first dialyzed overnight against the SEC buffer containing 0.5 mM TCEP and no EDTA in a Slide-A-Lyzer MINI Dialysis Device, 3.5 kDa MWCO. The dialyzed protein was quantitated by measuring UV absorbance and subjected to the deaminase assay as above in four modified buffer conditions, including (1) no added metal ions, (2) 20 μM ZnCl$_2$, (3) 1.0 mM MgCl$_2$, (4) 20 μM ZnCl$_2$ and 1.0 mM MgCl$_2$, with molar ratios between DddA(1290–1396) and DddA(1397–1422) of 1:1 and 1:10 (Extended Data Fig. 5). The Mg$^{2+}$-free reactions were supplemented with 1 mM MgCl$_2$ upon the addition of pfuEndoQ and heating to 60 °C.

### Reporting summary

Further information on research design is available in the Nature Portfolio Reporting Summary linked to this article.

## Data availability

Atomic coordinates and structure factors have been deposited in the Protein Data Bank (PDB) under accession codes 8E5E and 8E5D. Source data are provided with this paper. All other data are available from the authors upon request.

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

## Acknowledgements

This work was supported by grants from the US National Institutes of Health (NIGMS R35-GM118047 and NCI P01-CA234228 to H.A.). X-ray diffraction data were collected at the Northeastern Collaborative Access Team beamlines, which are funded by the US National Institutes of Health (NIGMS P30 GM124165). The Pilatus 6M detector on 24-ID-C beamline is funded by an NIH-ORIP HEI grant (S10 RR029205). This research used resources of the Advanced Photon Source, a US Department of Energy (DOE) Office of Science User Facility operated for the DOE Office of Science by Argonne National Laboratory under contract no. DE-AC02-06CH11357. We thank R. Harris for helpful suggestions.

## Author contributions

L.Y. performed protein purification, crystallization and biochemical analyses. K.S. performed crystallization, X-ray data collection and structure determination. H.A. managed the project and wrote the paper with input from all authors.

## Competing interests

The authors declare no competing interests.

## Additional information

**Extended data** is available for this paper at https://doi.org/10.1038/s41594-023-01034-3.

**Correspondence and requests for materials** should be addressed to Hideki Aihara.

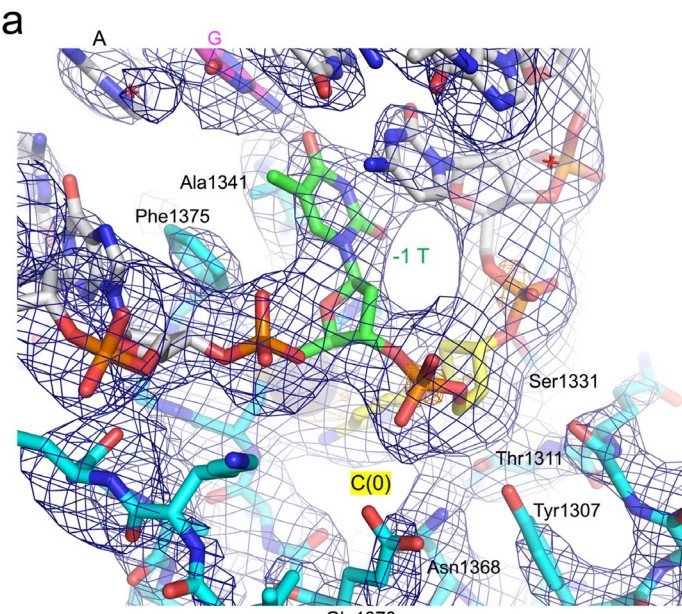

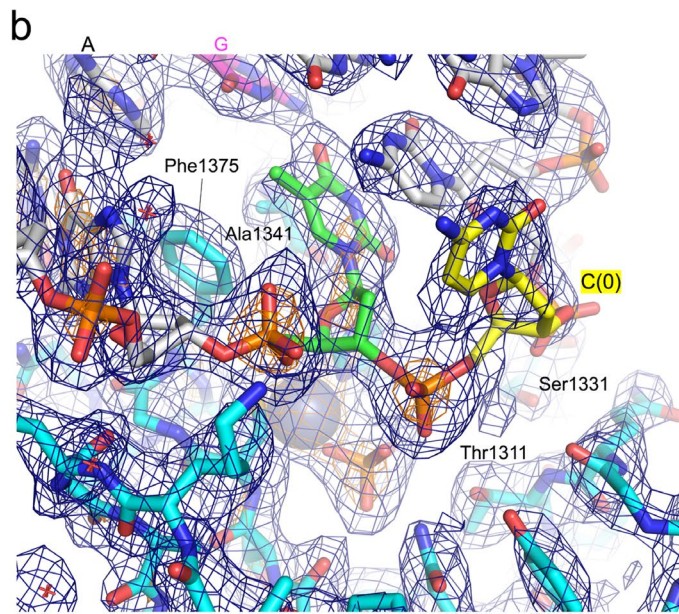

**Extended Data Fig. 1 | Alternative conformations of dsDNA bound to DddA.**
**a**, Structure with the target cytosine engaged in the active site pocket (8E5E). The 2mFo-DFc electron density map is contoured at 1.2 σ (blue) or 5.0 σ (orange) above the mean level. **b**, Structure with the target cytosine parked in the major groove (8E5D). The 2mFo-DFc electron density map is contoured at 1.5 σ (blue) or 5.0 σ (orange) above the mean level. The gray sphere represents the Zn ion in the active site. Red crosshairs represent water molecules.

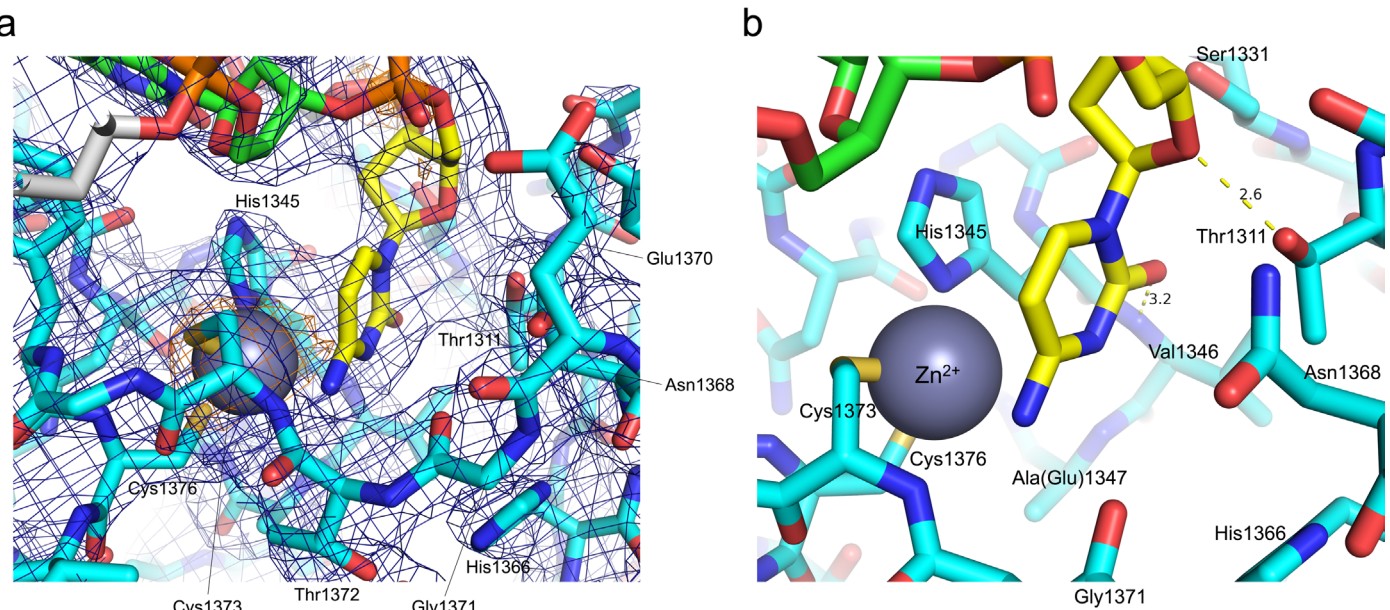

**Extended Data Fig. 2 | Close-up views of the target deoxycytidine in the active site pocket.** In **a**, 2mFo-DFc electron density map is contoured at 1.5 σ (blue) or 5.0 σ (orange) above the mean level. In **b**, hydrogen bonds between the nucleotide and surrounding protein residues are indicated by yellow dashed lines, with distances in angstrom.

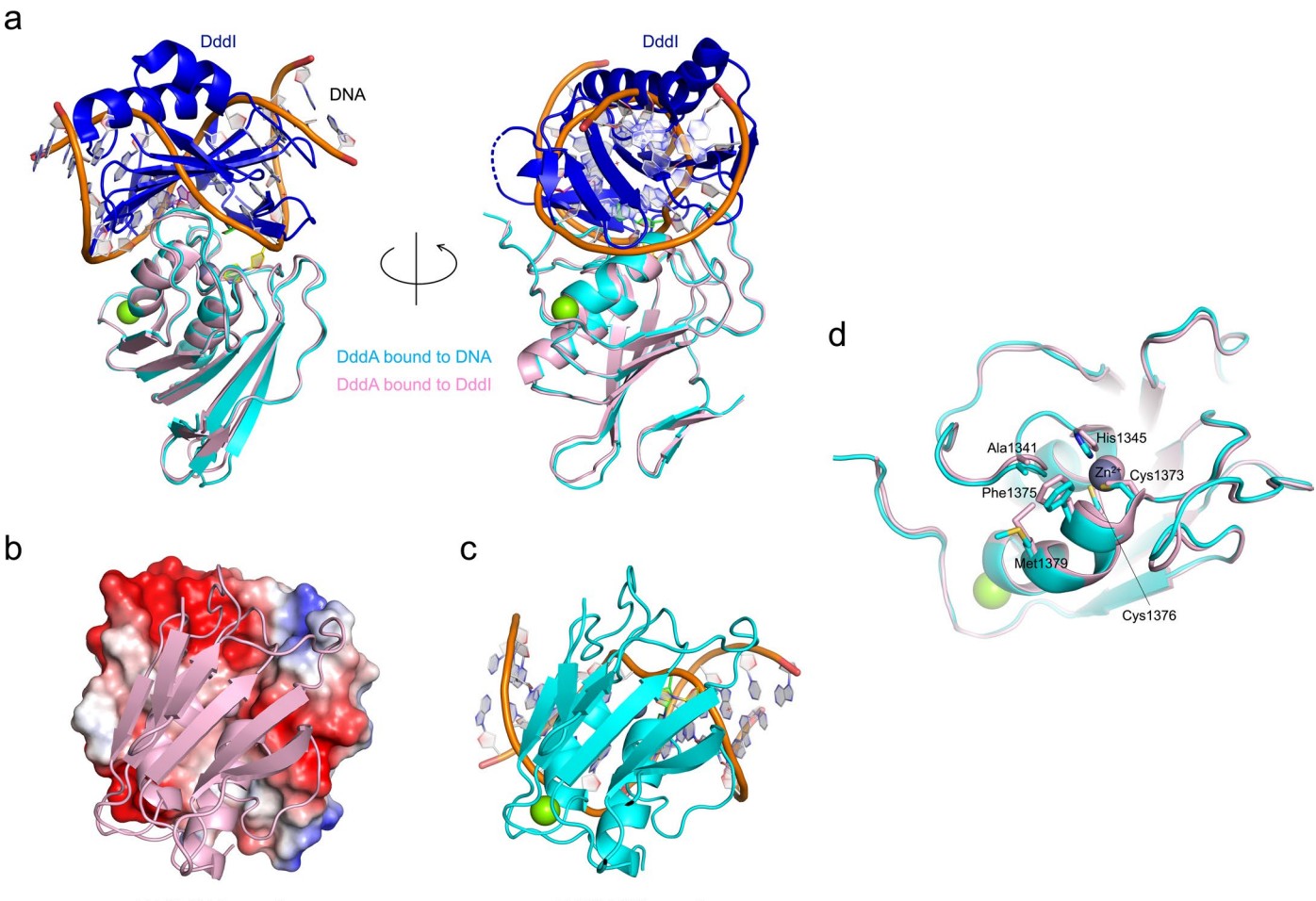

**Extended Data Fig. 3 | Comparison between DddI-bound and dsDNA-bound DddA structures. a**, Superposition of the dsDNA-bound DddA structure determined in this study and the previously reported DddI-bound DddA structure (PDB ID: 6u08)[12]. Note the complete overlap of DddI with DNA. **b**, DddA-DddI complex with DddA in the front. DddI is in the rear, with its surface colored according to electrostatic potential. **c**, DddA-dsDNA complex. Note that the distribution of negatively charged (red) patches in (**b**) matches DNA backbone positions in (**c**), suggesting DNA mimicry by DddI. **d**, Superposition between the DddI-bound and dsDNA-bound DddA structures, with the functionally important side chains highlighted.

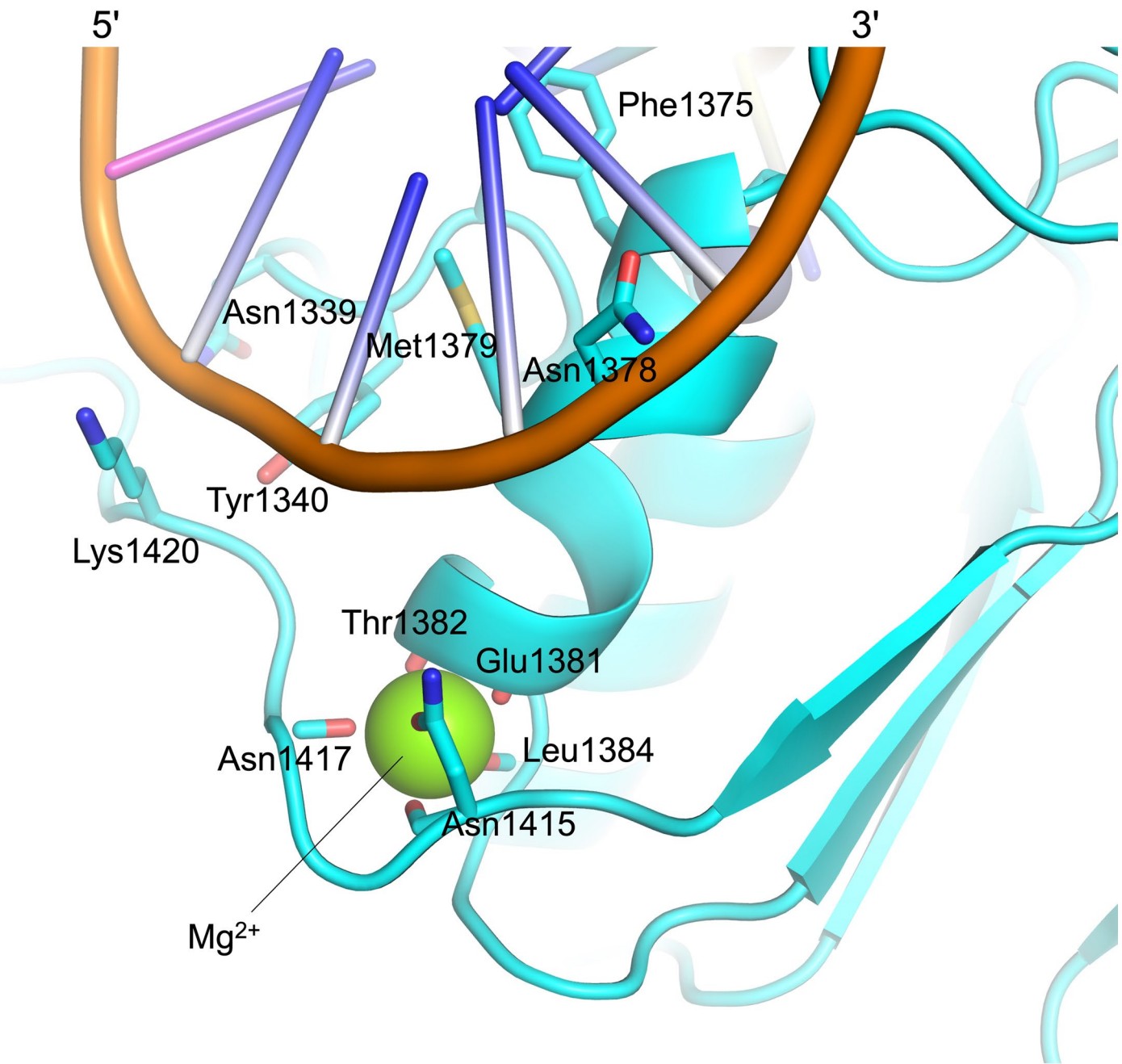

**Extended Data Fig. 4 | Magnesium ion coordination.** A Mg$^{2+}$ ion octahedrally coordinated by 5 main chain carbonyl and 1 side chain oxygen atoms, observed in both of our DddA-dsDNA complex structures.

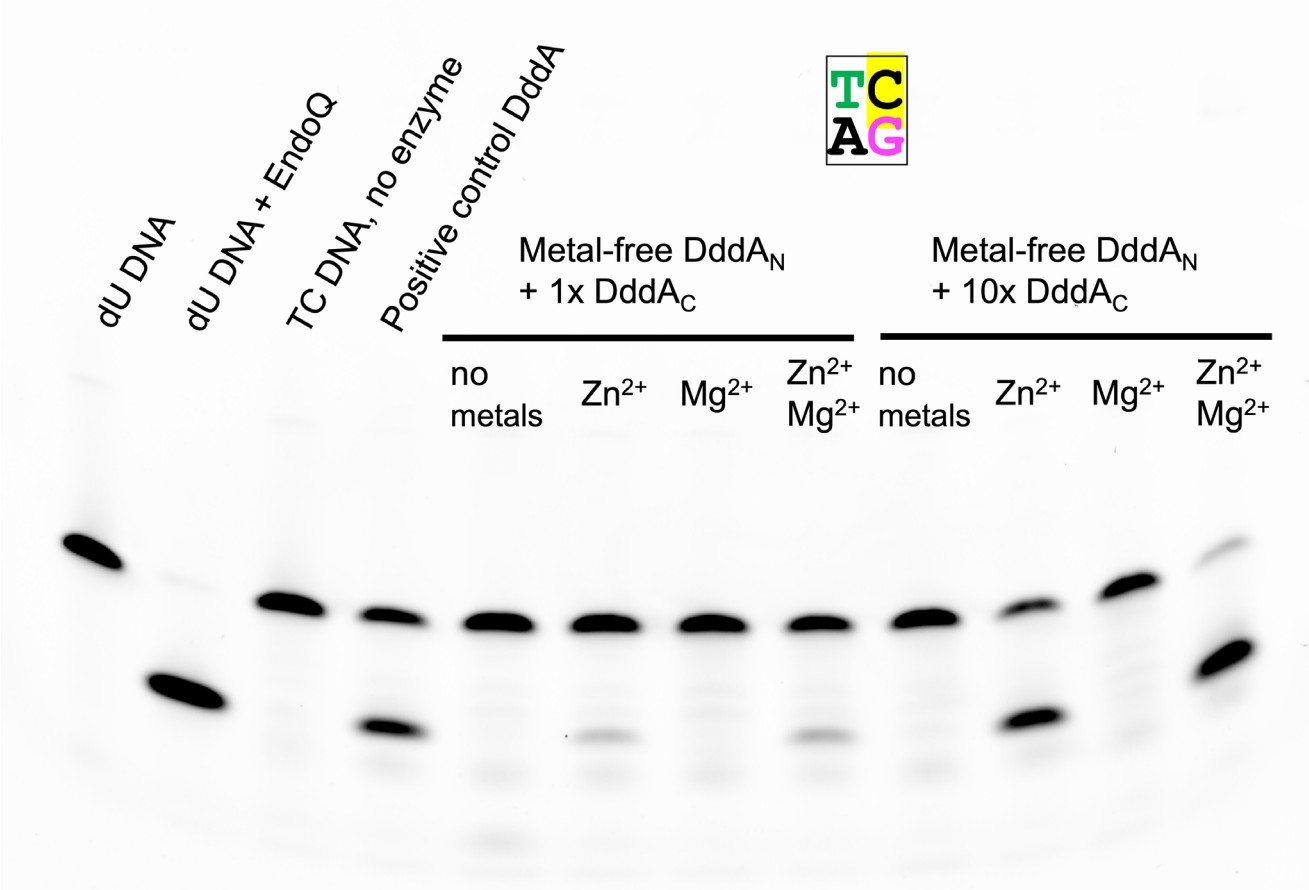

**Extended Data Fig. 5 | Metal ion dependency of DddA activity.** Deamination of the fully base-paired T**C**-containing dsDNA substrate by DddA in the absence of metal ions, and in the presence of $Zn^{2+}$, $Mg^{2+}$, or both $Zn^{2+}$ and $Mg^{2+}$. The activity was tested at two different molar ratios (1:1 and 1:10) between DddA(1290-1396) and DddA(1397-1422). The positive control reaction used DddA purified without EDTA in the standard assay condition as in the other experiments (*for example*, Figs. 4, 5). Representative result of two replicates is shown.

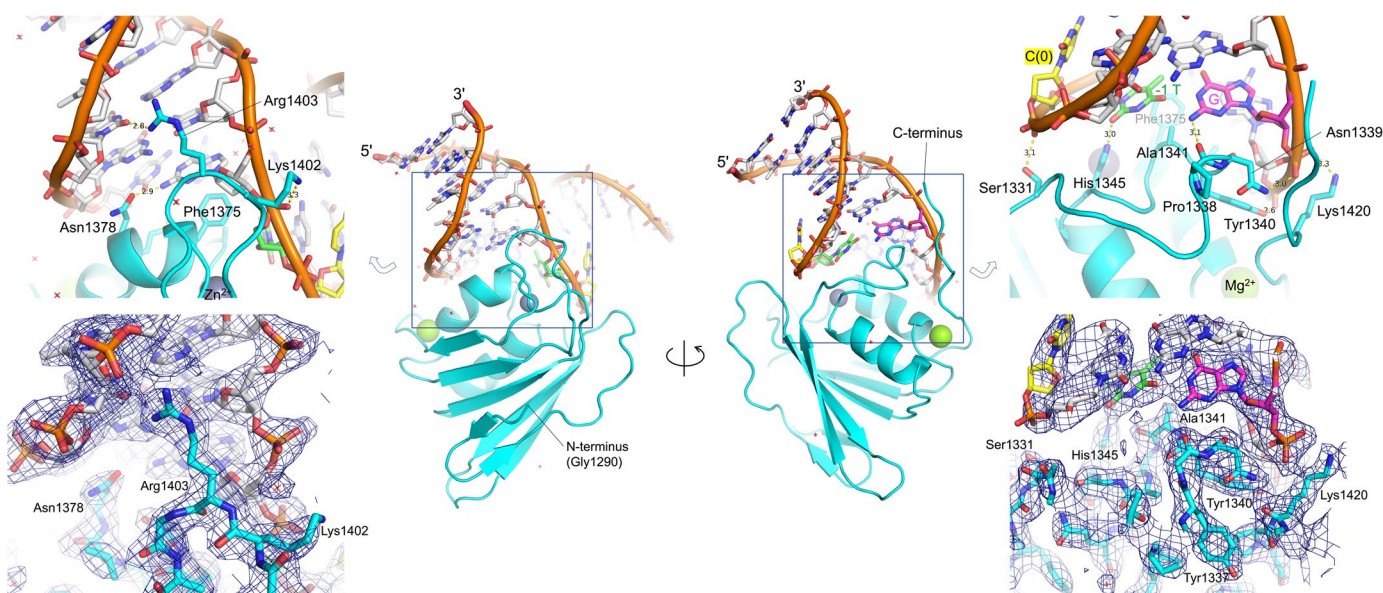

**Extended Data Fig. 6 | DddA-DNA contacts.** Zoomed views for the boxed regions of the DddA-dsDNA complex (the higher resolution 8E5D structure) are shown with hydrogen bonds and salt bridges indicated by dashed lines. The 2mFo-DFc electron density map is contoured at 1.5 σ (blue) or 5.0 σ (orange) above the mean level. The color scheme for nucleotides at the −1 and 0th positions follows that in Fig. 1a.

# 3' fluorescein-labeled substrates

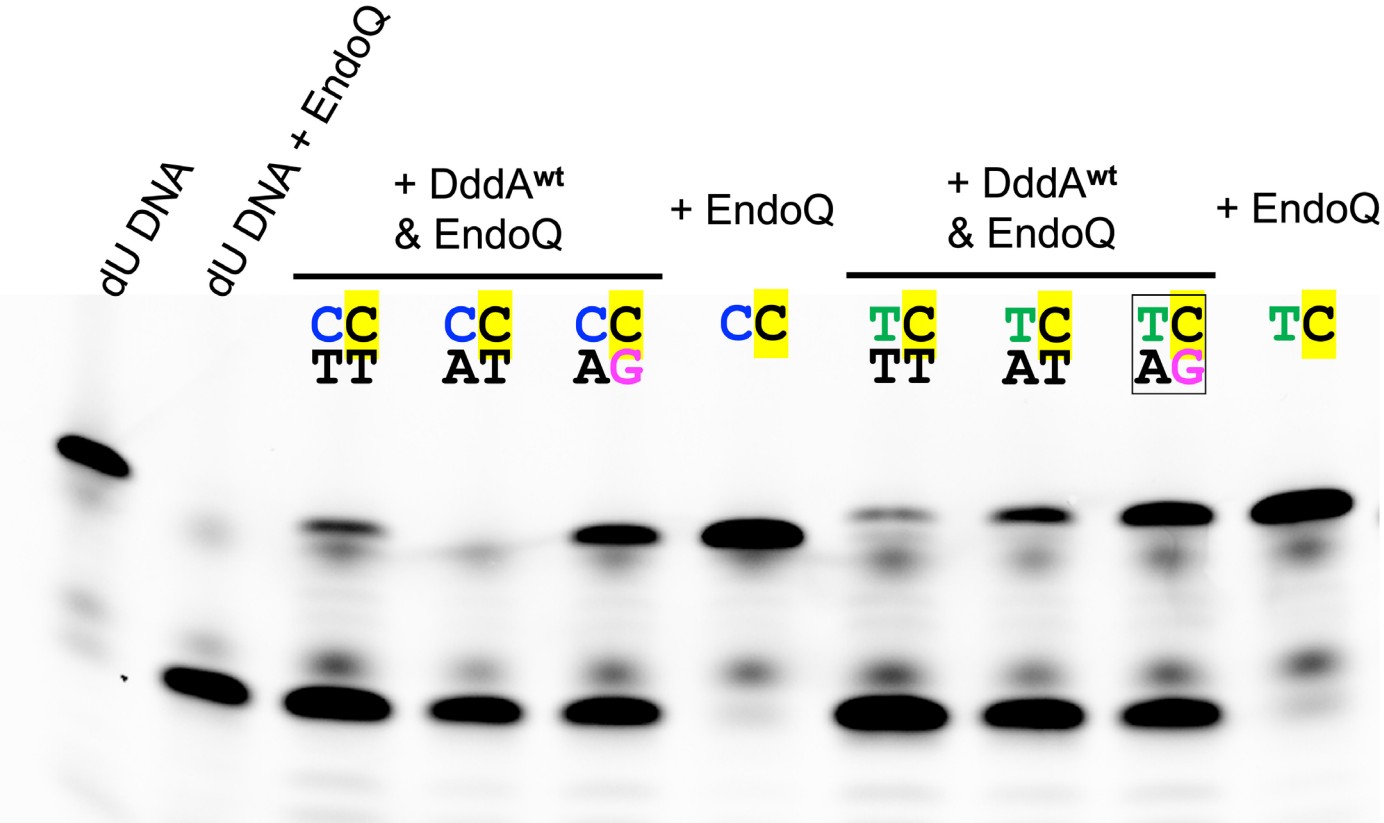

**Extended Data Fig. 7 | Deamination assay with 3' fluorescein-labeled DNA substrates.** The top and bottom strand sequences for the −1 and 0th positions are shown above each lane. Representative result of two replicates is shown.

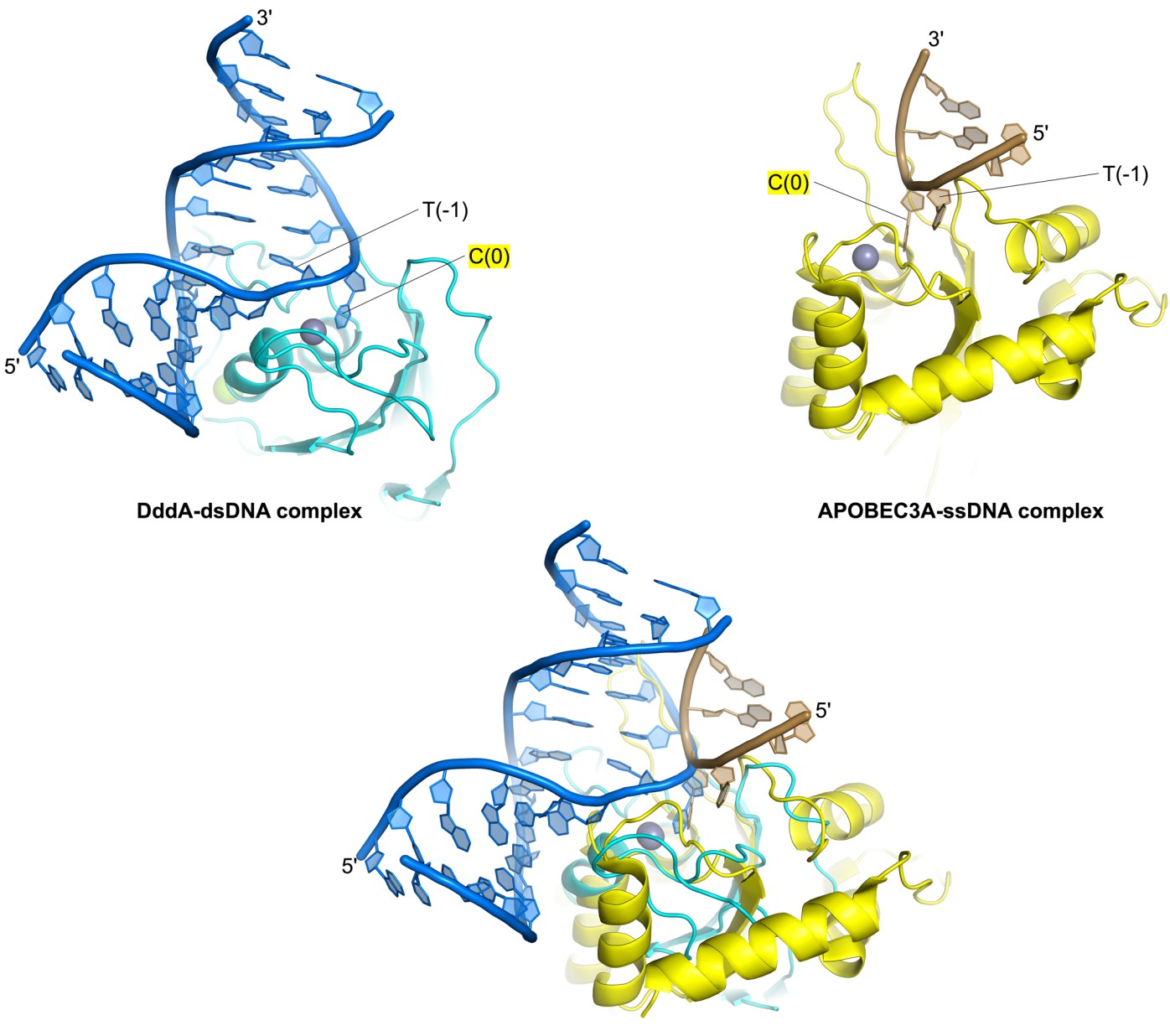

**DddA-dsDNA complex**

**APOBEC3A-ssDNA complex**

**Superimposed**

**Extended Data Fig. 8 | Distinct modes of TC-motif recognition by DddA and APOBEC3A (A3A).** A side-by-side comparison and a superposition between the DddA-dsDNA structure (this study) and the A3A-ssDNA complex structure[8], showing similar orientations of the target (0) cytosine and distinct positioning of the −1 thymine. The two complexes were aligned based on the conserved Zn-coordinating Cys and His residues and the catalytic Glu (mutated to Ala in both structures).

a

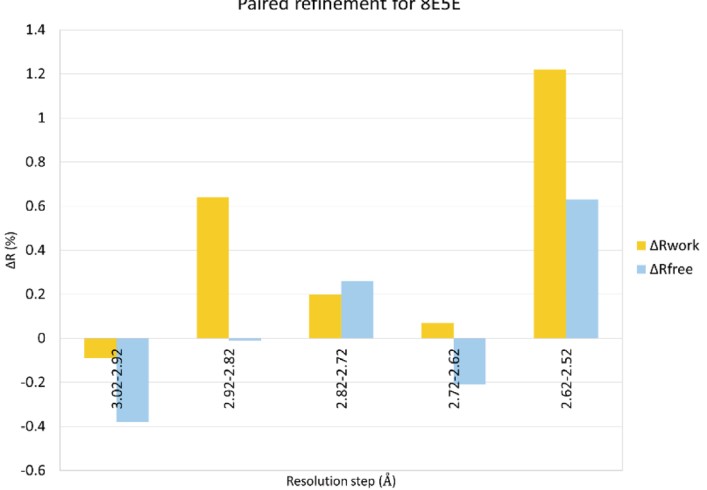

b

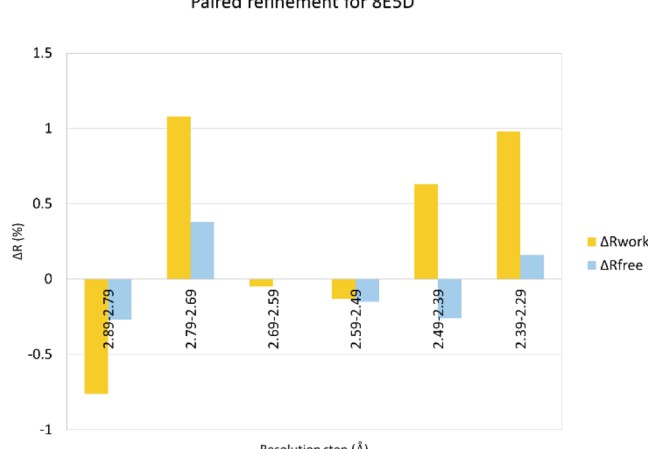

**Extended Data Fig. 9 | Paired refinement to assess the resolution limit.** Results of paired refinement[46] for DddA-dsDNA structure with the target cytosine engaged in the active site (**a**) and that with the target cytosine parked in the major groove (**b**). For each successive resolution step from low to high (x-axis), the pair of bars indicates the overall change in the $R_{work}$ (yellow) and $R_{free}$ (cyan) of the refined higher-resolution model compared to the refined lower-resolution model, with the two R values calculated against the lower-resolution data of the step.

# Reporting Summary

## Statistics

For all statistical analyses, confirm that the following items are present in the figure legend, table legend, main text, or Methods section.

| n/a | Confirmed | |
|---|---|---|
| ☒ | ☐ | The exact sample size (*n*) for each experimental group/condition, given as a discrete number and unit of measurement |
| ☒ | ☐ | A statement on whether measurements were taken from distinct samples or whether the same sample was measured repeatedly |
| ☒ | ☐ | The statistical test(s) used AND whether they are one- or two-sided<br>*Only common tests should be described solely by name; describe more complex techniques in the Methods section.* |
| ☒ | ☐ | A description of all covariates tested |
| ☒ | ☐ | A description of any assumptions or corrections, such as tests of normality and adjustment for multiple comparisons |
| ☒ | ☐ | A full description of the statistical parameters including central tendency (e.g. means) or other basic estimates (e.g. regression coefficient) AND variation (e.g. standard deviation) or associated estimates of uncertainty (e.g. confidence intervals) |
| ☒ | ☐ | For null hypothesis testing, the test statistic (e.g. *F*, *t*, *r*) with confidence intervals, effect sizes, degrees of freedom and *P* value noted<br>*Give P values as exact values whenever suitable.* |
| ☒ | ☐ | For Bayesian analysis, information on the choice of priors and Markov chain Monte Carlo settings |
| ☒ | ☐ | For hierarchical and complex designs, identification of the appropriate level for tests and full reporting of outcomes |
| ☒ | ☐ | Estimates of effect sizes (e.g. Cohen's *d*, Pearson's *r*), indicating how they were calculated |

*Our web collection on statistics for biologists contains articles on many of the points above.*

## Software and code

Policy information about availability of computer code

| Data collection | No software was used. |
|---|---|
| Data analysis | X-ray diffraction data were processed and analyzed using XDS (ver Feb 5, 2021), PHASER (ver 2.8.3), COOT (ver 0.94.1, 0.98.2), CCP4 (ver 8.0), and PHENIX (ver 1.19.2_4158, 1.20.1_4487). DNA groove width was calculated using CURVES+ (ver 2.6 / 2014). Structural figures were generated using PyMOL (ver 2.5.3). |

For manuscripts utilizing custom algorithms or software that are central to the research but not yet described in published literature, software must be made available to editors and reviewers. We strongly encourage code deposition in a community repository (e.g. GitHub). See the Nature Portfolio guidelines for submitting code & software for further information.

## Data

Policy information about availability of data

All manuscripts must include a data availability statement. This statement should provide the following information, where applicable:
- Accession codes, unique identifiers, or web links for publicly available datasets
- A description of any restrictions on data availability
- For clinical datasets or third party data, please ensure that the statement adheres to our policy

Atomic coordinates and structure factors have been deposited in the Protein Data Bank (PDB) under accession codes 8E5D and 8E5E. The PDB data (accession code 6U08) were used in the study.

## Human research participants

Policy information about studies involving human research participants and Sex and Gender in Research.

| | |
|---|---|
| Reporting on sex and gender | N/A |
| Population characteristics | N/A |
| Recruitment | N/A |
| Ethics oversight | N/A |

Note that full information on the approval of the study protocol must also be provided in the manuscript.

# Field-specific reporting

Please select the one below that is the best fit for your research. If you are not sure, read the appropriate sections before making your selection.

☒ Life sciences    ☐ Behavioural & social sciences    ☐ Ecological, evolutionary & environmental sciences

For a reference copy of the document with all sections, see nature.com/documents/nr-reporting-summary-flat.pdf

# Life sciences study design

All studies must disclose on these points even when the disclosure is negative.

| | |
|---|---|
| Sample size | No statistical methods were used to predetermine sample size. |
| Data exclusions | No data were excluded from the analyses. |
| Replication | All attempts at replication were successful. The numbers of replicated experiments are indicated in figure legends. |
| Randomization | Samples were not allocated to groups. |
| Blinding | Investigators were not blinded during data acquisition and analysis because it is not a common practice for the methods employed. |

# Reporting for specific materials, systems and methods

We require information from authors about some types of materials, experimental systems and methods used in many studies. Here, indicate whether each material, system or method listed is relevant to your study. If you are not sure if a list item applies to your research, read the appropriate section before selecting a response.

### Materials & experimental systems

| n/a | Involved in the study |
|---|---|
| ☒ | ☐ Antibodies |
| ☒ | ☐ Eukaryotic cell lines |
| ☒ | ☐ Palaeontology and archaeology |
| ☒ | ☐ Animals and other organisms |
| ☒ | ☐ Clinical data |
| ☒ | ☐ Dual use research of concern |

### Methods

| n/a | Involved in the study |
|---|---|
| ☒ | ☐ ChIP-seq |
| ☒ | ☐ Flow cytometry |
| ☒ | ☐ MRI-based neuroimaging |

