## [Peer Review File · Nature Structural & Molecular Biology]

Peer Review Information

Manuscript Title: Structural basis of sequence-specific cytosine deamination by double-stranded DNA deaminase toxin DddA

Corresponding author name(s): Hideki Aihara

Reviewer Comments & Decisions:

Decision Letter, initial version:
--

Message: 17th Jan 2023

Dear Dr. Aihara,

Thank you again for submitting your manuscript "Structural basis of sequence-specific cytosine deamination by double-stranded DNA deaminase toxin DddA". [I apologize for the delay in responding, which resulted from the difficulty in obtaining suitable referee reports. Nevertheless,] we now have comments (below) from the 3 reviewers who evaluated your paper. In light of those reports, we remain interested in your study and would like to see your response to the comments of the referees, in the form of a revised manuscript.

Referees 1 and 3 raised concerns that the crystallographic statistics aren't very strong so they suggest cutting the resolution off at 2.8-3.Å. Referee 2 recommended performing additional experiments showing that Mg coordination is required for DNA binding. Referees 1 and 2 requested a close up view of the critical interaction residues and bases.

Please be sure to address/respond to all concerns of the referees in full in a point-by-point response and highlight all changes in the revised manuscript text file. If you have comments that are intended for editors only, please include those in a separate cover letter.

We expect to see your revised manuscript within 6 weeks. If you cannot send it within this time, please contact us to discuss an extension; we would still consider your revision, provided that no similar work has been accepted for publication at NSMB or published elsewhere.

Reporting Summary:

When submitting the revised version of your manuscript, please pay close attention to our [href="https://www.nature.com/nature-portfolio/editorial-policies/image-integrity">Digital Image Integrity Guidelines. and to the following points below:](https://www.nature.com/nature-portfolio/editorial-policies/image-integrity)

Please note that all key data shown in the main figures as cropped gels or blots should be presented in uncropped form, with molecular weight markers. These data can be aggregated into a single supplementary figure item. While these data can be displayed in a relatively informal style, they must refer back to the relevant figures. These data should be submitted with the final revision, as source data, prior to acceptance, but you may want to start putting it together at this point.

Data availability: this journal strongly supports public availability of data. All data used in

accepted papers should be available via a public data repository, or alternatively, as Supplementary Information. If data can only be shared on request, please explain why in your Data Availability Statement, and also in the correspondence with your editor. Please note that for some data types, deposition in a public repository is mandatory - more information on our data deposition policies and available repositories can be found below: <https://www.nature.com/nature-research/editorial-policies/reporting-standards#availability-of-data>

[Redacted]

Sincerely,

Carolina Perdigoto, PhD
Chief Editor
Nature Structural & Molecular Biology
orcid.org/0000-0002-5783-7106

Reviewers' Comments:

Reviewer #1:

Remarks to the Author:

In the paper by Lin et al the authors present two crystal structures of Burkholderia cenocepacia DddA (catalytically inactive) in complex with a 14-bp dsDNA substrate containing the 5'-TC target sequence. These two structures give the first glimpse about how this base editing enzyme recognizes dsDNA, flipping out the substrate C0 for modification. The structures are complemented by extensive mutational, demonstrating that most of the interactions observed in the complex are essential (or at least strongly impacting) enzymatic function. This study significantly increases our understanding of how these dsDNA base editors recognize their substrates and is well presented.

The following questions should be addressed.

1. It is unclear if the first conformation of C0 in the complex is in a potential catalytic active conformation (or not) – none of the figures, really show the C0 in the context of the active site residues. This would be clarified with a zoomed in panel of the C0 and the active site. If it is not, then developing a potential model of the substrate complex with the C0 completely in the active site might be warranted in the discussion.
2. The potential significance of the 2nd conformation could be better delineated in Figure 2b and a discussion of how the structure might interconvert between the two structures. Additional panels c and d might show the DNA alone zoomed in t
3. The crystallographic statistics for both structures are worse than average structures of their resolutions as indicated by the pdb submission files – and the crystallographic data. Perhaps cutting the resolution off at 2.8-3Å to include only the highest quality data would be warranted.
4. For all the figures the depth cueing via shading should be removed, this makes it much more difficult for the structures to be completely visualized.

Reviewer #2:

Remarks to the Author:

In this manuscript authors described the crystal structure of DddA toxin domain complex with dsDNA substrate. Using the structure and biochemical experiments authors tried to elucidate the structural basis of DddA's sequence specificity for cytosine deamination at 5'-TC motif which was observed earlier. Authors also presented cytosine deamination activity results with different varying base sequence at -1 position and at the opposite base of target cytosine to support the observed domino effect the enzyme applies to flip the target cytosine to the active site pocket. Then they used mutational approach to confirm the role of residues involved in the process as observed from the structure. DddA is a new class of cytosine deaminase as it catalyzes deamination of cytosine to uracil in dsDNA unlike previously known other cytosine deaminases (CDA, AID, APOBEC etc) which do the same in ssDNA or RNA. Perhaps DddA is the first/only enzyme so far capable to deaminate cytosine to uracil in dsDNA and showed promising results and its advantages in base editing technology which has profound application in biomedical research and therapeutics. Considering the importance of DddA as a new base editing tool (CRISPR

independent) the structure of DddA with substrate DNA is highly valuable to understand the mechanism of the process and to develop future base editing tools with specific requirements. From these merits, I think this manuscript is worth to publish in Nature Structural & Molecular Biology after some improvement/modification.

Some questions/suggestions the authors may address in a revised manuscript:

1. In result, Overall Structure of DaaD-dsDNA Complex section, line 73-74, authors described "the target cytosine base is completely flipped out of the DNA double helix and captured in the active site pocket, where it interacts with the Zn ion (Fig. 2a)". From the current fig2a it is noticeable that target cytosine is flipped out of the helix and near the Zn ion. However, it is not clear how the Zn ion or other residues in the active site pocket interact with the flipped-out cytosine base to stabilize in the active site pocket. It will be clear if authors show another figure here showing details interactions of the target base with active site residues and Zn ion. Also, authors could consider a different angle to show the flipped-out cytosine in front unlike the current Fig. 2a which is barely visible for fogging.

2. In result, Overall Structure of DaaD-dsDNA Complex section, line 87-91, authors described "Besides the active site zinc ion, we observed electron density for a putative metal ion octahedrally coordinated by the backbone carbonyl oxygen of Glu1381, Thr1382, Leu1384, and Asn1417, and both the backbone and side chain oxygen atoms of Asn1415. This density was modeled as a magnesium ion, which appears to play a structural role to stabilize the DddA residues important for DNA-binding (Supplementary Fig. 1)."

From method section it is clear that, the only source of magnesium is the crystallization screening solution of crystal form#1 (8E5E), 0.2M magnesium chloride. The observation of Mg in crystal form#1 is possible. If authors modeled a magnesium at similar position in crystal form#2, what is the source of this Mg? Also, if authors claim this Mg coordination with corresponding residues structurally stabilize the DddA residues important for DNA-binding---Did authors perform any experiment to support this claim?

3. In result, Mechanism of TC motif recognition section, line 98-99, authors described "His1345, which is one of the Zn-coordinating residues, also donates a hydrogen-bond to the thymine O2 atom.

Does this interaction of His1345 with thymine O2 atom adjust the position of Zn/ other active site residues in active site pocket? Authors could compare with previously reported DNA free structure (PDB ID:6u08).

4. In result, Mechanism of TC motif recognition section, line 107-109, authors described "Upstream of the 5'-TC motif, Asn1378 and Arg1403 are inserted into the DNA minor groove and interact with bases at positions -2 to -4, respectively, which may modestly contribute to sequence preferences (Supplementary Fig. 2). Viewing supplemental Fig.2, it is difficult to follow all the interactions described in the text. It would be better to show more closeup view covering the interacting residues and bases with hydrogen bond interactions. Also, in left panel of supplemental Fig. 2, Arg1403 apparently interacting with opposite strand base of the target cytosine although the text describes position -4 base of same strand of target cytosine. This inconsistency needs to be fixed.

5. In discussion section, line 180-184, authors described "However, the mechanism of base flipping by DddA is distinct in that the intercalated phenylalanine replaces the adjacent (-1) thymine rather than the target (0) cytosine base itself (Fig. 3a). This unique

arrangement causes a “domino effect” and a shift in the register of base-pairing, with the target cytosine base extruded from the double helix.” Is there any evidence of this type of unique arrangement previously reported in the literature? If reported, please cite those references.

6. In discussion section, line 187-195, authors discuss about how DddA use the distinct mechanism from ssDNA cytosine deaminase A3A/A3B to recognize same 5'-TC target motif. It will be more visible to readers if authors present a figure superimposing two structures (DddA and A3a/A3B) highlighting the different/similar orientation and interaction of -1T. Although, -1T recognition is different, target (0) C recognition and catalytic interaction might be similar once it flipped out to the active site pocket in DddA. In fact, a previous report (Reference 12) highlighted the similarity of DddA with APOBEC family enzymes, as stated “Structure based homology searches revealed APOBEC enzymes as the closest structural relatives of DddAtox.” So, a discussion about the similarity/dissimilarity of DddA with A3A or A3B at target cytosine recognition will make it more complete. Again, a similar figure highlighting target (0) C at active site will be helpful.

7. In Methods section, a more details description of methods will be helpful to other researchers to use these methods in their own research or to reproduce the results mentioned here. From this current version it might be difficult. For an example, in Protein Expression and Purification section “The expressed protein was purified using nickel affinity and size exclusion chromatography”. A stepwise detail (mentioning materials used and their source) will be more helpful. Similarly, in Crystallization and structure determination section – Description in details about crystallization sample preparation (mentioning the dialysis tube cutoff size, how sample was concentrated etc.) is important.

Reviewer #3:

Remarks to the Author:

The manuscript entitled “Structural basis of sequence-specific cytosine deamination by double-stranded DNA deaminase toxin DddA” by Yin et al., described the co-crystal structure of the inactive mutant of DddA deaminase domain with a dsDNA substrate. The results show how DddA binds to dsDNA and specifically targets TC motif on the substrate strand. Specifically, the structures show that DddA binds to the minor groove of the dsDNA to induce bending and deformation of the dsDNA centered around the TC motif. In doing so, DddA uses its hydrophobic residue to invade the helix to disrupt the 5'(-1)T-A pair by displacing the 5' (-1) T, which in turn displaces the target (0) C to form a non-canonical T-G base-pair with the G that was paired to target (0) C. This tandem displacement allows the target C to flip out to the active site of DddA for a deamination reaction. Their biochemical experiments show that DNA base mismatches enhance DddA deaminase activity and relax its sequence selectivity, possibly making it much easier for the target C to flip out for deamination. The paper is concise, and the points mentioned above are clearly presented, and the conclusion is well supported by the data.

Below are a few specific comments:

1). DddA is a domain of a bacterial toxin. It would be good to have a cartoon illustration showing where is DddA located in the entire toxin protein, and also other domain boundaries of the protein.

- 2). They showed an alternative structure in which the displaced target (0) C is not located inside the DddA active site. They interpret it as an intermediate state. If it can be captured by X-ray structure, it means it's pretty stable. The author should describe the difference in the interactions in the DddA-DNA between the C inside the active center with the C outside the active center, and discuss why such an intermediate exists.
- 3). The authors used the "domino" effect to describe the displacement from the T (-1) to the C (0). When I first see that, I expect to see a few more sequential displacements after that, which is not the case based on the description. If it's very local displacement, it may be better not to use the "domino" effect. Something like tandem displacement if it's only 2 to 3 residues displacement.
- 4). The abstract states "This "domino effect" mechanism allows DddA to locate the target cytosine without flipping it into the active site." This statement is inaccurate or confusing. The data shows that it is the DddA binding to the dsDNA that deforms the dsDNA and flips the target C out to the active center of DddA (for example, by inserting a Phe residue of the DddA into the double helix to displace the T(-1), etc.). The other structure with the target C flipping out away from the active site is also caused by DddA binding to the dsDNA substrate but is not the active form. This should be restated.
- 5). The H1345 interacting with the minor groove is interesting, as His has been observed to be used by many other proteins to interact with the minor groove of dsDNA because the minor groove is mostly negatively charged and His is positively charged. Maybe the author can discuss the similarity and differences of this interaction with other proteins (such as the IFN- β Enhanceosome (IRF3, IRF7)).
- 6). An overall structure comparison between the apo-DddA and DNA-DddA should be shown, with a close-up view of the active center to show the difference.
- 7). The high-resolution range seems to be pushed. The I/σ at the highest resolution bin is 0.6 for both structures, and the CC1/2 is below 50%. Such data at the highest resolution bin at this redundancy of 5-7 as they have shown here will not contribute to the phasing at the resolution cut-off. Considering other statistics they show in the table, it's better to cut back the high resolution to the bin with I/σ above 1.0 and CC1/2 above 50%.
- 8) Fig. 1, panel c
It would be helpful to label the base "A" of the complementary strand (the unpaired A).
- 9) Supplementary Fig. 1
Label 5' and 3' of the DNA strand.
- 10) Supplementary Fig. 2
Suggest describing the color code of the bases in the figure legend.
- 11) Fig. 3, panel b
Move "+1" next to the black C
- 12) It's helpful to provide some discussion on the difference between GC/AG (poor substrate) and AC/AG (good substrate).

Author Rebuttal to Initial comments

We thank all reviewers for constructive comments and suggestions. Our point-by-point responses are shown below in blue.

Reviewers' Comments:

Reviewer #1:

Remarks to the Author:

In the paper by Lin et al the authors present two crystal structures of *Burkholderia cenocepacia* DddA (catalytically inactive) in complex with a 14-bp dsDNA substrate containing the 5'-TC target sequence. These two structures give the first glimpse about how this base editing enzyme recognizes dsDNA, flipping out the substrate C0 for modification. The structures are complemented by extensive mutational, demonstrating that most of the interactions observed in the complex are essential (or at least strongly impacting) enzymatic function. This study significantly increases our understanding of how these dsDNA base editors recognize their substrates and is well presented.

The following questions should be addressed.

1. It is unclear if the first conformation of C0 in the complex is in a potential catalytic active conformation (or not) – none of the figures, really show the C0 in the context of the active site residues. This would be clarified with a zoomed in panel of the C0 and the active site. If it is not, then developing a potential model of the substrate complex with the C0 completely in the active site might be warranted in the discussion.

We have added a new figure (Extended Data Fig. 2) that shows zoomed-in views of C(0) in the active site pocket. This is likely to be close to the catalytically relevant conformation, although it may not be in a strict sense, because we are missing the catalytic residue Glu1347 and (possibly as a consequence) there does not appear to be enough space for the nucleophilic water between the zinc ion and C4 position of the cytosine. Regardless, since the catalytic mechanism (chemistry) of Zn-dependent cytosine deamination is well-established based on earlier high-resolution studies (*e.g.*, Ref #24), we prefer not to discuss too much about the catalytic mechanism in this paper.

2. The potential significance of the 2nd conformation could be better delineated in Figure 2b and a discussion of how the structure might interconvert between the two structures. Additional panels c and d might show the DNA alone zoomed in t

We updated Fig. 2 to better depict the two distinct DNA conformations and their interaction with surrounding protein residues. Superposition of the two DNA conformations is shown in a new panel (Fig. 2c) to provide a sense of how the structure might interconvert between the two states.

3. The crystallographic statistics for both structures are worse than average structures of their resolutions as indicated by the pdb submission files – and the crystallographic data. Perhaps cutting the resolution off at 2.8-3Å to include only the highest quality data would be warranted.

We carefully re-processed the X-ray diffraction data and used the paired refinement procedure to re-evaluate the resolution limits (please see the updated Method section and Extended Data Fig. 9). Perhaps more important than the numerical cutoff values, we added several figures (Extended Data Figs. 1, 2, and 6) showing various regions of the electron density map to indicate how well the structural details are resolved.

4. For all the figures the depth cueing via shading should be removed, this makes it much more difficult for the structures to be completely visualized.

We agree that shading was too heavy in many of the figures and fixed this issue.

Reviewer #2:

Remarks to the Author:

In this manuscript authors described the crystal structure of DddA toxin domain complex with dsDNA substrate. Using the structure and biochemical experiments authors tried to elucidate the structural basis of DddA's sequence specificity for cytosine deamination at 5'-TC motif which was observed earlier. Authors also presented cytosine deamination activity results with different varying base sequence at -1 position and at the opposite base of target cytosine to support the observed domino effect the enzyme applies to flip the target cytosine to the active site pocket. Then they used mutational approach to confirm the role of residues involved in the process as observed from the structure. DddA is a new class of cytosine deaminase as it catalyzes deamination of cytosine to uracil in dsDNA unlike previously known other cytosine deaminases (CDA, AID, APOBEC etc) which do the same in ssDNA or RNA. Perhaps DddA is the first/only enzyme so far capable to deaminate cytosine to uracil in dsDNA and showed promising results and its advantages in base editing technology which has profound application in biomedical research and therapeutics. Considering the importance of DddA as a new base editing tool (CRISPR independent) the structure of DddA with substrate DNA is highly valuable to understand the mechanism of the process and to develop future base editing tools with specific requirements. From these merits, I think this manuscript is worth to publish in Nature Structural & Molecular Biology after some improvement/modification.

Some questions/suggestions the authors may address in a revised manuscript:

1. In result, Overall Structure of DaaD-dsDNA Complex section, line 73-74, authors described “the target cytosine base is completely flipped out of the DNA double helix and captured in the active site pocket, where it interacts with the Zn ion (Fig. 2a)”. From the current fig2a it is noticeable that target cytosine is flipped out of the helix and near the Zn ion. However, it is not clear how the Zn ion or other residues in the active site pocket interact with the flipped-out cytosine base to stabilize in the active site pocket. It will be clear if authors show another figure here showing details interactions of the target base with active site residues and Zn ion. Also, authors could consider a different angle to show the flipped-out cytosine in front unlike the current Fig. 2a which is barely visible for fogging.

We have added a new figure (Extended Data Fig. 2) that shows zoomed-in views of C(0) in the active site pocket and its interaction with surrounding residues. We also updated Fig. 2 to show the flipped-out cytosine base more clearly.

2. In result, Overall Structure of DaaD-dsDNA Complex section, line 87-91, authors described “Besides the active site zinc ion, we observed electron density for a putative metal ion octahedrally coordinated by the backbone carbonyl oxygen of Glu1381, Thr1382, Leu1384, and Asn1417, and both the backbone and side chain oxygen atoms of Asn1415. This density was modeled as a magnesium ion, which appears to play a structural role to stabilize the DddA residues important for DNA-binding (Supplementary Fig. 1).”

From method section it is clear that, the only source of magnesium is the crystallization screening solution of crystal form#1 (8E5E), 0.2M magnesium chloride. The observation of Mg in crystal form#1 is possible. If authors modeled a magnesium at similar position in crystal form#2, what is the source of this Mg? Also, if authors claim this Mg coordination with corresponding residues structurally stabilize the DddA residues important for DNA-binding--- Did authors perform any experiment to support this claim?

We observed electron density for the magnesium ion in both structures. Thus, we suspect that Mg^{2+} bound to DddA in *E. coli* expression remained bound during purification. To test whether the bound Mg^{2+} contributes to the deaminase activity, we purified DddA in the presence of 1 mM EDTA to strip bound Zn^{2+} and Mg^{2+} and then conducted the DNA deaminase assay with or without added metals. We found that Zn^{2+} is essential, as expected. Mg^{2+} turned out to be not essential, but it modestly increased the activity of DddA. We show these new data in Extended Data Fig. 5.

3. In result, Mechanism of TC motif recognition section, line 98-99, authors described “His1345, which is one of the Zn-coordinating residues, also donates a hydrogen-bond to the thymine O2 atom.

Does this interaction of His1345 with thymine O2 atom adjust the position of Zn/ other active site residues in active site pocket? Authors could compare with previously reported DNA free

structure (PDB ID:6u08).

A structural comparison between the DddI-bound and dsDNA-bound DddA is now shown in Extended Data Fig. 3. The positioning of the zinc ion or catalytically important residues does not change notably.

4. In result, Mechanism of TC motif recognition section, line 107-109, authors described “Upstream of the 5'-TC motif, Asn1378 and Arg1403 are inserted into the DNA minor groove and interact with bases at positions -2 to -4 , respectively, which may modestly contribute to sequence preferences (Supplementary Fig. 2). Viewing supplemental Fig.2, it is difficult to follow all the interactions described in the text. It would be better to show more closeup view covering the interacting residues and bases with hydrogen bond interactions. Also, in left panel of supplemental Fig. 2, Arg1403 apparently interacting with opposite strand base of the target cytosine although the text describes position -4 base of same strand of target cytosine. This inconsistency needs to be fixed.

We updated the figure (now Extended Data Fig. 6) to show closer views of these interactions. Arg1403 is indeed hydrogen-bonded to a thymine base from the non-deaminated strand. We clarified this in the text.

5. In discussion section, line 180-184, authors described “However, the mechanism of base flipping by DddA is distinct in that the intercalated phenylalanine replaces the adjacent (-1) thymine rather than the target (0) cytosine base itself (Fig. 3a). This unique arrangement causes a “domino effect” and a shift in the register of base-pairing, with the target cytosine base extruded from the double helix.” Is there any evident of this type of unique arrangement previously reported in the literature? If reported, please cite those references.

We are not aware of this type of arrangement previously reported in the literature.

6. In discussion section, line 187-195, authors discuss about how DddA use the distinct mechanism from ssDNA cytosine deaminase A3A/A3B to recognize same 5'-TC target motif. It will be more visible to readers if authors present a figure superimposing two structures (DddA and A3a/A3B) highlighting the different/similar orientation and interaction of -1 T. Although, -1 T recognition is different, target (0) C recognition and catalytic interaction might be similar once it flipped out to the active site pocket in DddA. In fact, a previous report (Reference 12) highlighted the similarity of DddA with APOBEC family enzymes, as stated “Structure based homology searches revealed APOBEC enzymes as the closest structural relatives of DddAtox.” So, a discussion about the similarity/dissimilarity of DddA with A3A or A3B at target cytosine recognition will make it more complete. Again, a similar figure highlighting target (0) C at active site will be helpful.

We added a new figure showing a superposition of A3A-ssDNA and DddA-dsDNA complexes to highlight their distinct modes of the TC motif recognition (Extended Data Fig. 8).

7. In Methods section, a more details description of methods will be helpful to other researchers to use these methods in their own research or to reproduce the results mentioned here. From this current version it might be difficult. For an example, in Protein Expression and Purification section “The expressed protein was purified using nickel affinity and size exclusion chromatography”. A stepwise detail (mentioning materials used and their source) will be more helpful. Similarly, in Crystallization and structure determination section – Description in details about crystallization sample preparation (mentioning the dialysis tube cutoff size, how sample was concentrated etc.) is important.

We updated the Methods section to include more details about protein expression/purification, crystallization, and X-ray data processing/analysis.

Reviewer #3:

Remarks to the Author:

The manuscript entitled “Structural basis of sequence-specific cytosine deamination by double-stranded DNA deaminase toxin DddA” by Yin et al., described the co-crystal structure of the inactive mutant of DddA deaminase domain with a dsDNA substrate. The results show how DddA binds to dsDNA and specifically targets TC motif on the substrate strand. Specifically, the structures show that DddA binds to the minor groove of the dsDNA to induce bending and deformation of the dsDNA centered around the TC motif. In doing so, DddA uses its hydrophobic residue to invade the helix to disrupt the 5'(-1)T-A pair by displacing the 5' (-1) T, which in turn displaces the target (0) C to form a non-canonical T-G base-pair with the G that was paired to target (0) C. This tandem displacement allows the target C to flip out to the active site of DddA for a deamination reaction. Their biochemical experiments show that DNA base mismatches enhance DddA deaminase activity and relax its sequence selectivity, possibly making it much easier for the target C to flip out for deamination. The paper is concise, and the points mentioned above are clearly presented, and the conclusion is well supported by the data.

Below are a few specific comments:

1). DddA is a domain of a bacterial toxin. It would be good to have a cartoon illustration showing where is DddA located in the entire toxin protein, and also other domain boundaries of the protein.

A cartoon diagram has been added to Fig. 1 to show where the deaminase toxin domain of DddA is relative to the full-length protein. We are not including the other putative motifs of DddA shown in a figure from the original Mok et al. paper (PAAR, proline-alanine-alanine-arginine; RHS, rearrangement hotspot), because they are not relevant to our studies here.

2). They showed an alternative structure in which the displaced target (0) C is not located inside the DddA active site. They interpret it as an intermediate state. If it can be captured by X-ray structure, it means it's pretty stable. The author should describe the difference in the interactions in the DddA-DNA between the C inside the active center with the C outside the active center, and discuss why such an intermediate exists.

We have updated Fig. 2 and added Extended Data Fig. 1 to better illustrate a comparison between the two structures. We don't necessarily understand the significance of the alternative conformation, but one possibility is that it represents how DddA scans through a DNA sequence to locate target cytosines without flipping them into the active site pocket. We added a brief mentioning about this possibility in Discussion.

3). The authors used the "domino" effect to describe the displacement from the T (-1) to the C (0). When I first see that, I expect to see a few more sequential displacements after that, which is not the case based on the description. If it's very local displacement, it may be better not to use the "domino" effect. Something like tandem displacement if it's only 2 to 3 residues displacement.

"Tandem displacement" sounds more fitting indeed. We replaced 'domino effect' with 'tandem displacement'.

4). The abstract states "This "domino effect" mechanism allows DddA to locate the target cytosine without flipping it into the active site." This statement is inaccurate or confusing. The data shows that it is the DddA binding to the dsDNA that deforms the dsDNA and flips the target C out to the active center of DddA (for example, by inserting a Phe residue of the DddA into the double helix to displace the T(-1), etc.). The other structure with the target C flipping out away from the active site is also caused by DddA binding to the dsDNA substrate but is not the active form. This should be restated.

We think that the unique structure of dsDNA bound to DddA featuring the T-G pair enables DddA to locate TC motifs in dsDNA without directly sensing the cytosine base in the active site pocket.

5). The H1345 interacting with the minor groove is interesting, as His has been observed to be used by many other proteins to interact with the minor groove of dsDNA because the minor groove is mostly negatively charged and His is positively charged. Maybe the author can discuss the similarity and differences of this interaction with other proteins (such as the IFN- β Enhanceosome (IRF3, IRF7)).

This is an interesting point. We added a sentence to mention that the interaction made by DddA His1345 in the widened minor groove is distinct from the DNA shape readout by His insertion into a compressed DNA minor groove as observed for other proteins such as IRF3/7.

6). An overall structure comparison between the apo-DddA and DNA-DddA should be shown, with a close-up view of the active center to show the difference.

A comparison between the DddI-bound and DNA-bound DddA structures is shown in a new figure (Extended Data Fig. 3).

7). The high-resolution range seems to be pushed. The I/σ at the highest resolution bin is 0.6 for both structures, and the CC1/2 is below 50%. Such data at the highest resolution bin at this redundancy of 5-7 as they have shown here will not contribute to the phasing at the resolution cut-off. Considering other statistics they show in the table, it's better to cut back the high resolution to the bin with I/σ above 1.0 and CC1/2 above 50%.

We carefully re-processed the X-ray diffraction data and used the paired refinement procedure to re-evaluate the resolution limits (please see the updated Method section and Extended Data Fig. 9). Perhaps more important than the numerical cutoff values, we added several figures (Extended Data Figs. 1, 2, and 6) showing various regions of the electron density map to indicate how well the structural details are resolved.

8) Fig. 1, panel c

It would be helpful to label the base "A" of the complementary strand (the unpaired A).

We added a label for the unpaired A.

9) Supplementary Fig. 1

Label 5' and 3' of the DNA strand.

We added labels for 5' and 3' of the DNA strand.

10) Supplementary Fig. 2

Suggest describing the color code of the bases in the figure legend.

We added the following sentence to the figure legend (now Extended Data Fig. 6):

"The color scheme for nucleotides at the -1 and 0th positions follows that in Fig. 1a"

11) Fig. 3, panel b

Move "+1" next to the black C

The position of “+1” has been adjusted.

12) It's helpful to provide some discussion on the difference between GC/AG (poor substrate) and AC/AG (good substrate).

We added a brief statement about the sequence preference, as following. We prefer not to discuss more specifics because we don't really understand the basis of the observation.

“Residual sequence dependence observed for the mismatched substrates (e.g., Fig. 4c, lane 9 vs. 11) may reflect how efficiently the –1 base replaces the target (0) cytosine by interacting with its juxtaposed base and the surrounding protein residues, including His1345, in the distorted dsDNA conformation.”

Decision Letter, first revision:

Message: Our ref: NSMB-A46809A

12th Apr 2023

Dear Dr. Aihara,

Thank you for submitting your revised manuscript "Structural basis of sequence-specific cytosine deamination by double-stranded DNA deaminase toxin DddA" (NSMB-A46809A). It has now been seen by the original referees and their comments are below. The reviewers find that the paper has improved in revision, and therefore we'll be happy to accept it in principle in Nature Structural & Molecular Biology, pending minor revisions to satisfy the referees' final requests and to comply with our editorial and formatting guidelines.

To facilitate our work at this stage, it is important that we have a copy of the main text as a word file. If you could please send along a word version of this file as soon as possible, we would greatly appreciate it; please make sure to copy the NSMB account (cc'ed above).

Sincerely,

Dimitris Typas
Associate Editor

Nature Structural & Molecular Biology
ORCID: 0000-0002-8737-1319

Reviewer #1 (Remarks to the Author):

The updated figures and electron density greatly increased the clarity of this manuscript and alleviated the concerns I previously had.

Reviewer #2 (Remarks to the Author):

The authors made a good faith effort to address my concerns. Now the methods are in more details and figures represent more clear views. I recommend acceptance of the manuscript.

Reviewer #3 (Remarks to the Author):

The authors have adequately addressed the concerns.

Final Decision Letter:

Message 12th Jun 2023

:

Dear Dr. Aihara,

We are now happy to accept your revised paper "Structural basis of sequence-specific cytosine deamination by double-stranded DNA deaminase toxin DddA" for publication as a Article in Nature Structural & Molecular Biology.

Due to the importance of these deadlines, we ask that you please let us know now whether

you will be difficult to contact over the next month. If this is the case, we ask you provide us with the contact information (email, phone and fax) of someone who will be able to check the proofs on your behalf, and who will be available to address any last-minute problems.

Your paper will be published online soon after we receive proof corrections and will appear in print in the next available issue. You can find out your date of online publication by contacting the production team shortly after sending your proof corrections. Content is published online weekly on Mondays and Thursdays, and the embargo is set at 16:00 London time (GMT)/11:00 am US Eastern time (EST) on the day of publication. Now is the time to inform your Public Relations or Press Office about your paper, as they might be interested in promoting its publication. This will allow them time to prepare an accurate and satisfactory press release. Include your manuscript tracking number (NSMB-A46809B) and our journal name, which they will need when they contact our press office.

About one week before your paper is published online, we shall be distributing a press release to news organizations worldwide, which may very well include details of your work. We are happy for your institution or funding agency to prepare its own press release, but it must mention the embargo date and Nature Structural & Molecular Biology. If you or your Press Office have any enquiries in the meantime, please contact press@nature.com.

An online order form for reprints of your paper is available at http://www.nature.com/reprints

<https://www.nature.com/reprints/author-reprints.html>. Please let your coauthors and your institutions' public affairs office know that they are also welcome to order reprints by this method.

Please note that *Nature Structural & Molecular Biology* is a Transformative Journal (TJ). Authors may publish their research with us through the traditional subscription access route or make their paper immediately open access through payment of an article-processing charge (APC). Authors will not be required to make a final decision about access to their article until it has been accepted. [Find out more about Transformative Journals](https://www.springernature.com/gp/open-research/transformative-journals)

[Find out more about Transformative Journals](https://www.springernature.com/gp/open-research/transformative-journals)

Authors may need to take specific actions to achieve [compliance with funder and institutional open access mandates](https://www.springernature.com/gp/open-research/funding/policy-compliance-faqs). If your research is supported by a funder that requires immediate open access (e.g. according to [Plan S principles](https://www.springernature.com/gp/open-research/plan-s-compliance)) then you should select the gold OA route, and we will direct you to the compliant route where possible. For authors selecting the subscription publication route, the journal's standard licensing terms will need to be accepted, including [self-archiving policies](https://www.springernature.com/gp/open-research/policies/journal-policies). Those licensing terms will supersede any other terms that the author or any third party may assert apply to any version of the manuscript.

Sincerely,

Dimitris Typas
Associate Editor
Nature Structural & Molecular Biology
ORCID: 0000-0002-8737-1319

Click here if you would like to recommend Nature Structural & Molecular Biology to your librarian:

<http://www.nature.com/subscriptions/recommend.html#forms>